# Metanetworks as Regulatory Operators: Learning to Edit for Requirement Compliance

## Abstract

As machine learning models are increasingly deployed in high-stakes settings, e.g. as decision support systems in various societal sectors or in critical infrastructure, designers and auditors are facing the need to ensure that models satisfy a wider variety of requirements (e.g. compliance with regulations, fairness, computational constraints) beyond performance. Although most of them are the subject of ongoing studies, typical approaches face critical challenges: post-processing methods tend to compromise performance, which is often counteracted by fine-tuning or, worse, training from scratch, an often time-consuming or even unavailable strategy. This raises the following question: *"Can we efficiently edit models to satisfy requirements, without sacrificing their utility?"* In this work, we approach this with a unifying framework, in a data-driven manner, i.e. we *learn to edit* neural networks (NNs), where the editor is an NN itself - a graph metanetwork - and editing amounts to a single inference step. In particular, the metanetwork is trained on NN populations to minimise an objective consisting of two terms: the requirement to be enforced and the preservation of the NN's utility. We experiment with diverse tasks (the data minimisation principle, bias mitigation and weight pruning) improving the trade-offs between performance, requirement satisfaction and time efficiency compared to popular post-processing or re-training alternatives.

## 1 Introduction

How to ensure machine learning models (ML) are unbiased, i.e. they do not *discriminate* based on sensitive attributes, such as race or gender? How to prevent them from *memorising* personal data? How to certify they are *robust* against adversarial attacks or in safety-critical applications? How to avoid *overusing resources*, such as energy (sustainability) or data (privacy)?

Driven by the growing and widespread deployment of AI algorithms, such as Neural Networks (NNs) in everyday life, these are only a few of the pressing questions that are posed by stakeholders, such as scholars, professionals, regulators, citizens and end-users. Generally put, there is a call to move beyond plain *performance/accuracy*, which has been the primary objective so far, (Díaz-Rodríguez et al., 2023; Liu et al., 2021) and design models that adhere to one or more additional requirements related to the task at hand, e.g. not violating rights, minimising environmental costs, keeping their decisions within safety boundaries, not reproducing unfounded information, etc.

These diverse requirements share a critical characteristic: they often emerge *after* the models are trained and deployed. New regulations are enacted (e.g. the EU AI Act), vulnerabilities are discovered post-deployment, or deployment contexts change unexpectedly. This temporal mismatch between model training and requirement specification creates a fundamental challenge that can be summarised into a shared framing: **Machine learning models need to be made compliant with various requirements without compromising their intended behaviour.**

Current approaches to this challenge face significant limitations. Post-processing methods often severely compromise performance, which is typically counteracted by fine-tuning or, worse, training from scratch—strategies that are time-consuming, computationally expensive, and often unavailable due to data privacy or intellectual property constraints.

We posit that this challenge should be tackled by taking into account two important considerations:

(1) Requirements are diverse and currently expanding. Therefore, developing requirement-specific methods, e.g. as has been done for model debiasing (Hardt et al., 2016) and pruning (Frankle & Carbin, 2019), can be limiting. Instead, we propose a unified framework by formulating a given requirement as a mathematical objective involving the model's parameters and/or outputs (for instance, minimising computational requirements via weight pruning can be written as $\min \|\boldsymbol{\theta}\|_0$, where $\boldsymbol{\theta}$ are the vectorised weights). Thereafter, we arrive at a *multi-objective* problem, where the objectives are the requirements as well as a metric measuring deviations from the intended model behaviour. A single objective comprising a weighted sum of the terms is used for optimisation.

(2) Ensuring requirement compliance should be efficient and flexible. Oftentimes, one or more requirements might not be present during the training of the model, e.g. when a new regulation is adopted or when a vulnerability was not anticipated. However, solving the multi-objective problem for every new requirement is time and resource-intensive. In contrast to a large body of ML literature that deals with multi-objective problems (Kendall et al., 2018; Sener & Koltun, 2018; Chen et al., 2018; Lin et al., 2019; Navon et al., 2021), we aim to circumvent the optimisation (be it training or fine-tuning) altogether, and *edit* models in a post-hoc fashion. In other words, we seek to identify a map from initial model parameters to edited ones that are requirement-compliant.

We approach this problem in a data-driven manner, capitalising on the recent advancements in *weight space learning* (Schürholt et al., 2025). In particular, we train a *metanetwork*, i.e. a specialised NN (equivariant to parameter symmetries), to edit the parameters of other NNs. We do so in an unsupervised manner, using NN parameter populations and optimising an estimate of the weighted objective. Crucially, once trained, the metanetwork can edit any model from the same task distribution in a *single forward pass*, making compliance achievable in seconds rather than hours or days.

This work establishes a foundational framework for learned model editing, opening a new research direction at the intersection of weight space learning, regulatory compliance, and efficient model adaptation. Here, we demonstrate our approach on three requirements using MLPs as a proof of concept; however, our framework is designed to catalyse future research into universal model editors and practical tools for regulatory compliance in deployed AI systems. Our contributions are as follows:

- We provide a unifying mathematical framework for NN requirement compliance using a multi-objective optimisation formulation.
- We devise a methodology to solve this problem efficiently and flexibly using a *learnable NN editing* paradigm, implemented with the recently introduced metanetworks.
- We specify our methodology on three requirements: data minimisation, fairness, and computational efficiency via weight pruning, formulating them mathematically.
- Our method is evaluated on diverse tasks, demonstrating consistent improvements over post-processing and retraining baselines.

## 2 RELATED WORK

**NN requirements & AI Auditing.** The need for auditing AI algorithms, in particular NNs, i.e. assessing and ensuring that they behave as intended and comply to certain standards, has emerged from the acknowledgement that automated systems display unwanted behaviours and perpetuate or even amplify societal biases and harms (Buolamwini & Gebru, 2018; Angwin et al., 2022). Additionally, legal mandates for AI auditing have proliferated across major jurisdictions (European Union, 2016; 2021; 2022; U.S. Executive Office of the President, 2023; New York City Council, 2021; , TC260; Government of Canada, 2022), with notable examples the EU GDPR and AI Act.

Trained models are stress-tested either by their designers (Ganguli et al., 2022) or by external auditors (Raji et al., 2022) to identify undesired behaviours. Typical areas that are investigated are societal implications, e.g. biases (Caton & Haas, 2024; Huszár et al., 2022), copyright infringements (Somepalli et al., 2023) and environmental concerns (Lacoste et al., 2019), transparency and explainability (Ribeiro et al., 2016), and robustness Carlini & Wagner (2017).

**NN Editing.** Nonetheless, these findings are rarely actionable, i.e. they do not provide insights on how to rectify the operation of NNs. The field that studies the latter is known as NN editing and was

initially approached with retraining/fine-tuning methods. However, as models grow increasingly complex, the need to modify them without retraining has become paramount (Mitchell et al., 2022; Meng et al., 2022a;b). In the context of compliance with requirements, typical cases include the following. *Fairness* post-processing methods (Hardt et al., 2016; Pleiss et al., 2017; Alghamdi et al., 2022; Chen et al., 2024) manipulate weights to debias model predictions. *Model compression* techniques, such as pruning (Frankle & Carbin, 2019; Han et al., 2016) and quantisation (Jacob et al., 2018), aim to reduce model size and improve efficiency while preserving performance. *Unlearning* techniques (Bourtoule et al., 2021) identify and remove the influence of specific training examples from learned parameters - critical for privacy compliance and addressing data quality issues.

Yet, these methods still require extra processing power and are application-specific. Instead, in our work, we propose an efficient, general-purpose alternative that produces NN edits at a single step. Our method can be used by designers, as well as auditors, provided that white-box access is given (access to model parameters), an important desideratum discussed in Casper et al. (2024).

**Weight Space Learning - Metanetworks.** Motivated by the abundant publicly available trained models and fuelled by the potential impact of the proposed applications, the emerging field of weight space learning Schürholt et al. (2025) - data-driven methodologies that process NN parameters - has gained significant traction over the last years. Initial efforts (Unterthiner et al., 2020; Eilertsen et al., 2020; Schürholt et al., 2021; 2022), apply standard NNs either to vectorised parameters or their statistics, while a series of works (Xu et al., 2022; Luigi et al., 2023; Dupont et al., 2022; Bauer et al., 2023) have focused on the particular case of Implicit Neural Representations (Sitzmann et al., 2020).

In contrast to the above, a recent stream of works has dominated the field, focusing on *equivariant metanetworks* that account for parameter space symmetries. These include the works of Navon et al. (2023) and Zhou et al. (2023a;b) that focus on permutation symmetries for MLPs and CNNs, which have been extended to more general architectures by Zhou et al. (2024); Lim et al. (2024); Kofinas et al. (2024). Recently, other types of symmetries have been studied, such as scaling (Kalogeropoulos et al., 2024; Tran et al., 2024; Vo et al., 2025), those present in Transformers (Tran et al., 2025), Low-Rank Adapter weights (Putterman et al., 2025) and NN gradients (Gelberg et al., 2025). Finally, research in this field has extended beyond metanetwork design to study a variety of related problems (Schürholt et al., 2024; Shamsian et al., 2024; Kahana et al., 2024; Zhao et al., 2022; Erkoç et al., 2023). In this work, we employ the general *graph metanetwork* paradigm of Lim et al. (2024), where the NN is modelled as a graph and then processed by a Graph Neural Network (GNN).

## 3 REQUIREMENT COMPLIANCE: MULTI-OBJECTIVE FORMULATION

**Notation.** In the following sections, vectors and matrices will be denoted with bold-face letters, e.g. $\mathbf{x}, \mathbf{X}$ and sets with calligraphic $\mathcal{X}$. A normal font notation will be used for miscellaneous purposes (mostly indices, functions and distributions). Datapoint (input) functions will be denoted with $f$, while functions of parameters will be denoted with fraktur font $\mathfrak{F}$.

**Problem Statement.** Let $f_{G,\boldsymbol{\theta}} : \mathcal{X} \to \mathcal{Y}$ be a model (typically a NN), parameterised by a computational graph $G \in \mathcal{G}$ and a vector of (learnable) parameters $\boldsymbol{\theta} \in \Theta$. We use $\mathcal{X}, \mathcal{Y}$ and $\mathcal{G}, \Theta$ to denote the input and output spaces and the spaces of computational graphs and parameters, respectively. Additionally, denote with $p_\mathrm{d}$ the distribution from which input-output pairs $(\mathbf{x}, y)$ are sampled, where $\mathbf{x} \in \mathcal{X}, y \in \mathcal{Y}$. The model $f_{G,\boldsymbol{\theta}}$ is subject to an editing procedure aiming to make it compliant with one or more requirements, while preserving its behaviour as much as possible. To translate this into mathematical statements, we aim to produce a new NN parameter pair $G', \boldsymbol{\theta}'$ that optimises the following pair of objective functions:

- **Preservation Objective:** Ensure that the original and the new model will have similar output when sampling datapoints from the data distribution. Formally:

$$\min_{G', \boldsymbol{\theta}'} d\Big( f_{G,\boldsymbol{\theta}}, f_{G',\boldsymbol{\theta}'}, p_\mathrm{d} \Big), \tag{1}$$

where $d(\cdot, \cdot, \cdot)$ measures the distance between the two functions.[1]

---

[1] In the case of untrained models, this term could be reformulated to reflect accuracy optimisation. However, for this paper, we will deal only with the case of trained models that need to be edited for compliance.

- **Requirement Objective:** Ensures that the new model will behave as required. Formally:

$$\min_{G', \boldsymbol{\theta}'} r\Big(G', \boldsymbol{\theta}', p_{\mathrm{d}}\Big), \qquad (2)$$

where $r(\cdot, \cdot, \cdot)$ is a function that encapsulates all the requirements imposed.

Simultaneously optimising for the above results in the following multi-objective formulation:

$$\min_{G', \boldsymbol{\theta}'} \Big( \underbrace{d\Big(f_{G, \boldsymbol{\theta}}, f_{G', \boldsymbol{\theta}'}, p_{\mathrm{d}}\Big)}_{\text{preservation objective}}, \; \underbrace{r\Big(G', \boldsymbol{\theta}', p_{\mathrm{d}}\Big)}_{\text{requirement objective}} \Big). \qquad (3)$$

## 3.1 Preservation Objective

A plethora of metrics can be used to compare model outputs, depending on the structure of the output space $\mathcal{Y}$. In the common case where $f$ is a classifier, the model outputs a vector of classification probabilities for each class, i.e. $\mathcal{Y}$ is a probability simplex, and therefore the comparison should be done via a probability metric. Throughout the paper, and without loss of generality, we employ the *Jensen–Shannon divergence*, a symmetric and bounded similarity measure between distributions, and compute its expectation over $p_{\mathrm{d}}$ (see appendix A.3.2).

## 3.2 Requirement Objective

Now, let us focus on certain examples of the requirement objective. We aim to cover a broad range of categories. We select examples from: (1) *Regulatory compliance*, in particular, the data minimisation principle, which is encountered in most data protection regulations, such as the EU GDPR European Union (2016). (2) *Non-violation of rights*, in particular the right to non-discrimination, which translates to algorithmic fairness metrics in mathematical terms. (3) *Computational efficiency*, achieved via weight pruning (NN sparsity).

### 3.2.1 Case 1: Data Minimization Principle

Data Minimisation (DM) mandates that only the necessary information for the task at hand is stored and processed. However, in the context of ML models, it is unknown which of the input features are required for a model's decision. Furthermore, it is likely that the model uses features of lesser importance as shortcuts to make decisions, an unwanted behaviour as per the DM principle. Hence, it is unclear how to enforce or verify that DM is respected when performing algorithmic auditing.

Given the above, first and foremost, it is necessary to express DM rigorously as a requirement objective. The most straightforward strategy is to define a binary mask that deactivates the input features that should not be considered. Equivalently, we can deactivate the corresponding input nodes of the model, denoted with $\mathcal{V}_{\mathrm{in}}(G')$, where $G'$ is its computational graph. Formally, the requirement objective becomes:

$$r\Big(G', \boldsymbol{\theta}', p_{\mathrm{d}}\Big) = |\mathcal{V}_{\mathrm{in}}(G')|. \qquad (4)$$

Observe that this requirement depends only on the model structure and not its outputs.

**Differentiability.** It is evident that eq. (4) is non-differentiable (it is a node count, a discrete variable), which prevents gradient-based optimisation. We therefore re-express it using the binary mask formulation described above, in a way that permits performing a continuous relaxation. In particular, to edit the model, we maintain the original computational graph and mask the outgoing weights of the deactivated input nodes via an auxiliary (differentiable) function $\mu \colon (G', \boldsymbol{\theta}') = (G, \mu(\tilde{\boldsymbol{\theta}}, \mathbf{m}))$, where $\mathbf{m} = [\mathbf{m}_1, \ldots, \mathbf{m}_{|\mathcal{V}_{\mathrm{in}}(G)|}]$ is the mask vector, i.e. $\mathbf{m}_i \in [0, 1]$ is a variable corresponding to input node $i$ that indicates its activation/deactivation, and $\tilde{\boldsymbol{\theta}}$ are preliminary edited parameters before masking. More information can be found in appendix A.1.1. Formally, the multi-objective becomes:

$$\min_{\mathbf{m}, \tilde{\boldsymbol{\theta}}} \Big( \mathop{\mathbb{E}}_{(\mathbf{x}, y) \sim p_{\mathrm{d}}} \Big[ \mathrm{JSD}\Big( f_{G, \boldsymbol{\theta}}(\mathbf{x}), f_{G, \mu(\tilde{\boldsymbol{\theta}}, \mathbf{m})}(\mathbf{x}) \Big) \Big], \; \sum_{i=1}^{|\mathcal{V}_{\mathrm{in}}(G)|} \mathbf{m}_i \Big), \qquad (5)$$

### 3.2.2 CASE 2: FAIRNESS

With an increasing number of ML models being deployed for decision-making, it is crucial to ensure they do not exhibit discriminatory behaviour. In computer science, multiple algorithmic fairness criteria have been proposed to enforce this, such as demographic parity (Kamiran & Calders, 2009), equal opportunity (Hardt et al., 2016) and counterfactual fairness (Kusner et al., 2017). Without loss of generality, we focus on the *equalised odds (EO)* criterion (Hardt et al., 2016). EO seeks to ensure that a model's prediction errors are distributed equally across different groups (as partitioned by e.g. gender or race), matching the true positive rate (*TPR*) and false positive rate (*FPR*) for different groups. Formally, for a set $\mathcal{S}$ of demographic groups and a set of $\mathcal{K}$ classes, EO is defined as:

$$\text{TPR}_{i,k}(f_{G,\boldsymbol{\theta}}, p_\mathrm{d}) = \text{TPR}_{j,k}(f_{G,\boldsymbol{\theta}}, p_\mathrm{d}), \quad \text{FPR}_{i,k}(f_{G,\boldsymbol{\theta}}, p_\mathrm{d}) = \text{FPR}_{j,k}(f_{G,\boldsymbol{\theta}}, p_\mathrm{d}), \ \forall i,j \in \mathcal{S}, \forall k \in \mathcal{K}. \tag{6}$$

and the rates are given by:

$$\text{TPR}_{i,k}(f_{G,\boldsymbol{\theta}}, p_\mathrm{d}) = \mathop{\mathbb{P}}_{(\mathbf{x},y,s)\sim p_\mathrm{d}} (\hat{y}(\mathbf{x}) = k \mid y = k, s = i), \tag{7}$$

$$\text{FPR}_{i,k}(f_{G,\boldsymbol{\theta}}, p_\mathrm{d}) = \mathop{\mathbb{P}}_{(\mathbf{x},y,s)\sim p_\mathrm{d}} [\hat{y}(\mathbf{x}) = k \mid y \neq k, s = i], \tag{8}$$

with $\hat{y}(\mathbf{x}) = \operatorname{argmax}_k f_{G,\boldsymbol{\theta}}(\mathbf{x})$ the predicted class. However, the above represents a fairness criterion, i.e. it is either satisfied or not, rather than a quantitative metric. To transform it into a requirement objective, we use the metric known as *equalised odds difference* (Bellamy et al., 2018):

$$r\Big(G', \boldsymbol{\theta}', p_\mathrm{d}\Big) = \text{EOD}(f_{G',\boldsymbol{\theta}'}, p_\mathrm{d}) = \max_{\substack{i,j\in\mathcal{S},\, i<j \\ k\in\mathcal{K}}} \{\big|\text{TPR}_{i,k}(f_{G,\boldsymbol{\theta}}, p_\mathrm{d}) - \text{TPR}_{j,k}(f_{G,\boldsymbol{\theta}}, p_\mathrm{d})\big|, \tag{9}$$

$$\big|\text{FPR}_{i,k}(f_{G,\boldsymbol{\theta}}, p_\mathrm{d}) - \text{FPR}_{j,k}(f_{G,\boldsymbol{\theta}}, p_\mathrm{d})\big|\}$$

**Differentiability:** As shown in eq. (9), our requirement objective relies on the predicted hard labels of the edited model, which hinders the differentiability of our method. Similarly to DM, we use a continuous relaxation by applying softmax-with-temperature on the output logits of the edited model. More information can be found in appendix A.1.2.

### 3.2.3 CASE 3: WEIGHT PRUNING

To address compute and energy requirements, variable methods have been studied for model compression (Hinton et al., 2015; Han et al., 2015; Wu et al., 2016), with pruning emerging as one of the most prominent techniques (Yu et al., 2018; Wang et al., 2019). NN pruning involves removing redundant parameters, e.g. groups of parameters (Yu et al., 2018; Fang et al., 2023; Li et al., 2017) or individual weights (Dong et al., 2017), thereby reducing model size and potentially accelerating inference. In our experiments, we showcase our results on weight pruning. Let $\mathcal{E}(G')$ be the edge set of the NN and $G'$ its computational graph. Now the requirement objective becomes:

$$r\Big(G', \boldsymbol{\theta}', p_\mathrm{d}\Big) = |\mathcal{E}(G')|. \tag{10}$$

**Differentiability.** Observe the similarities of the above to the DM objective of eq. (4), which leads to the same differentiability issue. As before, we make the multi-objective amenable to a continuous relaxation using a binary mask formulation and an auxiliary differentiable function $\mu$ that masks out the deactivated weights:

$$\min_{\mathbf{m},\tilde{\boldsymbol{\theta}}} \left( \mathop{\mathbb{E}}_{(\mathbf{x},y)\sim p_\mathrm{d}} \Big[\text{JSD}\Big(f_{G,\boldsymbol{\theta}}(\mathbf{x}), f_{G,\mu(\tilde{\boldsymbol{\theta}},\mathbf{m})}(\mathbf{x})\Big)\Big], \sum_{i=1}^{|\mathcal{E}(G)|} \mathbf{m}_i \right). \tag{11}$$

Note that, although plain pruning alone entails simply removing redundant parameters, we also consider updating the remaining parameters, which is shown to improve the experimental results. Please refer to appendix A.1.3 for further details.

## 4 REQUIREMENT COMPLIANCE: LEARNING TO EDIT

A major hurdle becomes evident by inspecting eq. (3): for every model subject to editing and every new requirement, an optimisation problem needs to be solved from scratch. This undoubtedly entails

significant resource costs (time, computational, financial and environmental). To address this, we adopt an alternative perspective that unfolds in the following section.

Let $\mathcal{P}(G, \boldsymbol{\theta}, p_\mathrm{d})$ be the set of Pareto optimal solutions of eq. (3), i.e. the set of all solutions for which none of the objectives can be improved without deteriorating at least one of the other objectives. Further, assume a *scalarisation* of the multi-objective problem to a single objective. A typical choice is linear scalarisation (which guarantees that its solutions will be Pareto optimal) by using a weighting coefficient $\lambda > 0$ as follows:

$$\min_{G', \boldsymbol{\theta}'} d\Big( f_{G, \boldsymbol{\theta}}, f_{G', \boldsymbol{\theta}'}, p_\mathrm{d} \Big) + \lambda r\Big( G', \boldsymbol{\theta}', p_\mathrm{d} \Big), \tag{12}$$

where we assume that any solution in $\mathcal{P}(G, \boldsymbol{\theta}, p_\mathrm{d})$ can be reached for some $\lambda > 0$.[2] Now, similar to all parametric optimisation problems, eq. (12), for fixed $p_\mathrm{d}$ and $\lambda$, gives rise to a mapping $\mathfrak{F}^*$ from initial $(G, \boldsymbol{\theta})$ to edited $(G', \boldsymbol{\theta}')$ NN parameters, where the latter is a minimiser of eq. (12). We therefore define $\mathfrak{F}^* : \mathcal{G} \times \Theta \to \mathcal{G} \times \Theta$ as a function whose output is an arbitrary minimiser:

$$\mathfrak{F}^*(G, \boldsymbol{\theta}; \lambda, p_\mathrm{d}) \in \min_{G', \boldsymbol{\theta}'} d\Big( f_{G, \boldsymbol{\theta}}, f_{G', \boldsymbol{\theta}'}, p_\mathrm{d} \Big) + \lambda r\Big( G', \boldsymbol{\theta}', p_\mathrm{d} \Big). \tag{13}$$

This reframing lies at the heart of our approach: *We will replace the optimisation solver with a function that directly maps original to edited networks.* The familiar reader will observe that one can find this function using machine learning, i.e. by collecting a dataset of NN parameters and learning to approximate the mapping $\mathfrak{F}^*$ in a data-driven manner. In particular, assume the editor has access to a dataset of NN parameters sampled i.i.d. from a distribution $p_\mathrm{m}$ on $\mathcal{G} \times \Theta$. Additionally, denote with $\mathfrak{F}_\phi : \mathcal{G} \times \Theta \to \mathcal{G} \times \Theta$ a *metanetwork* parametrised by $\phi$. Then, we can approximate $\mathfrak{F}^*$ for fixed $\lambda, p_\mathrm{d}$ with $\mathfrak{F}_\phi$ by formulating the following *unsupervised learning objective*:

$$\min_{\phi \in \Phi} \mathbb{E}_{(G, \boldsymbol{\theta}) \sim p_\mathrm{m}} \Big[ d\Big( f_{G, \boldsymbol{\theta}}, f_{\mathfrak{F}_\phi(G, \boldsymbol{\theta})}, p_\mathrm{d} \Big) + \lambda r\Big( \mathfrak{F}_\phi(G, \boldsymbol{\theta}), p_\mathrm{d} \Big) \Big] \tag{14}$$

**Practical considerations: Symmetries.** Examining eq. (12), one may observe that for every $(G, \boldsymbol{\theta})$ there exists a set of $(G', \boldsymbol{\theta}')$ that are minimisers to the problem, rather than a single solution. This is possible for any optimisation problem, but in the case of NNs, we are certain due to the existence of *parameter symmetries*. In particular, several works (Hecht-Nielsen, 1990; Chen et al., 1993; Godfrey et al., 2022) have identified parameter transformations (e.g. hidden neuron permutations) that keep the model function $f$ and, in turn, the preservation objective of eq. (1) unaffected. This is also true for various requirements, including the ones we experiment with in this paper (see appendix A.1).

Therefore, accounting for parameter symmetries can significantly facilitate optimisation of eq. (12) and eq. (14). In the latter, this is what motivates us to employ *equivariant metanetworks*, and in particular a modified version of the graph metanetworks of Lim et al. (2024). This ensures that pairs of equivalent inputs $(G, \boldsymbol{\theta})$ will be mapped to pairs of equivalent outputs $(G, \boldsymbol{\theta}')$. Further details on our architecture for $\mathfrak{F}$ can be found in appendix A.2

**Practical considerations: Data.** An important question that arises is what data are required by the editor to train the metanetwork. First, *during training, the editor needs samples from $p_d$*, since the preservation and in certain cases (e.g. fairness) the requirement objectives depend on $p_\mathrm{d}$. However, these do not need to be the same as the ones used to train the NNs. In our experimental section, we use a different and significantly smaller set of datapoints and observe satisfactory results. This is an important finding, e.g. for the case that the editor is an external party, since the training data might be unavailable due to privacy or intellectual property concerns. Moreover, *for a fixed $p_d$, during inference, the editor does not need samples from $p_d$*, which further reinforces the argument above.

Second, *for a fixed $p_d$, during training, the editor needs samples from $p_m$, i.e. a dataset of NNs that solve the same task*, to approximate the outer expectation of eq. (14). Note that these do not necessarily need to be NNs with high accuracy, i.e. fully tuned models and as a result, they are easier to collect. In fact, in our experiments, we create our datasets by training NNs with various hyperparameters, so as to span a wide range of accuracy scores.

**Remark: Multi-task learnable editing.** So far, we have only considered the case of a fixed $p_\mathrm{d}$, i.e. learning to edit models that solve the same task. However, undoubtedly, it is easier to collect a

---

[2]This is not true when the Pareto front is not convex, as discussed in e.g. (Lin et al., 2019), but for simplicity, here we avoided more complicated multi-objective approaches.

dataset using NNs trained on diverse tasks (such repositories exist in e.g. Hugging Face). On the other hand, this is a significantly more complicated scenario, since now the metanetwork will have to be a function of the form $\mathfrak{F}(G, \boldsymbol{\theta}, p_\mathrm{d})$ - it should be able to process NN input/output pairs apart from NN parameters, not to mention that the NN distribution $p_\mathrm{m}$ will be considerably more diverse. To date, these problems have not been addressed in the metanetwork literature and deserve deeper and more careful examination, and are therefore left to future work.

## 5 EXPERIMENTS

**Experimental Setup.** We evaluate our method on three types of requirements on MLPs that process tabular data. We construct two comprehensive datasets of trained MLPs (NN populations) with varying architectures and hyperparameters using two widely-studied benchmarks from the UCI repository (Kelly et al., 2024): *Adult* and *Bank Marketing*. These serve as *representative real-world tabular datasets*. Following the notation established for model and data distributions ($p_\mathrm{m}$ and $p_\mathrm{d}$, respectively) in previous sections, we will be referring to the corresponding datasets as $\mathcal{D}_\mathrm{m}$ and $\mathcal{D}_\mathrm{d}$. Detailed information about the dataset construction process is provided in appendix A.6. In all our experiments, we train our metanetwork on the train split of $\mathcal{D}_\mathrm{m}$ using a subset of the validation split of $\mathcal{D}_\mathrm{d}$ (unseen samples during training of the original model) to compare on the function space. Finally, we plot the results on the test split of $\mathcal{D}_\mathrm{m}$ using the test split of $\mathcal{D}_\mathrm{d}$ for data samples. Further implementation details can be found in appendix A.4.

Since all our setups define a multi-objective problem, in all cases we visualise trade-off curves (Pareto front). On the $x$-axis, we place the average requirement objective and on the $y$-axis, the average JS Divergence, measuring how the edited model's underlying function has drifted from the original. Since we seek to minimise both objectives, each point $\mathbf{p}_i \in \mathbb{R}^2$ dominates a point $\mathbf{p}_j \in \mathbb{R}^2$ if $\mathbf{p}_i[k] \leq \mathbf{p}_j[k], \forall k \in \{1, 2\}$ and $\mathbf{p}_i[k] < \mathbf{p}_j[k]$, for at least one $k \in \{1, 2\}$. Points that are not dominated by any other point form the Pareto front, representing the optimal trade-offs between the two competing objectives. We conduct a hyperparameter search, including the $\lambda$ values, and select the models that lie on the Pareto front of the validation set. Finally, we evaluate only the selected points on the test split and use them to construct the figures presented in our experiments.

Evidently, predicting parameter edits at inference time provides our method with significant time-efficiency advantages. To quantitatively assess this benefit relative to the baselines in each experiment, we conduct evaluations under a controlled environment using the $\mathcal{D}_\mathrm{m}$ of the Adult dataset. We report the results in table 1.

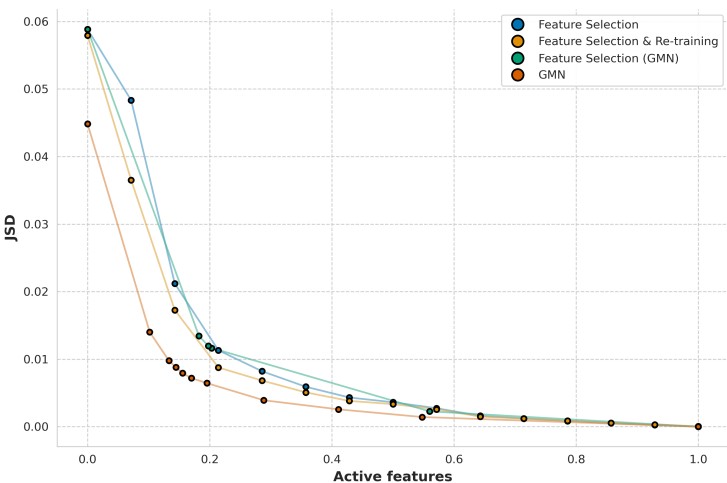

Figure 1: Data Minimization on the Adult dataset.

**Data Minimization.** We establish baselines using feature importance methods. First, we compute Permutation Feature Importance (PFI) Breiman (2001) to rank input features by their relevance for each dataset. For the naive *FS* (feature selection) baseline, we directly feed the masked input to the

original model. For the *FS & Retrain* baseline, we apply a knowledge distillation approach Hinton et al. (2015), treating the original model as the teacher and training a student model that receives only masked inputs. The training objective uses Jensen-Shannon (JS) divergence (Lin, 2002) as the loss function. This knowledge distillation baseline requires substantial computational resources, as each experiment is conducted independently for every model in the test split of $\mathcal{D}_m$. Finally, in *FS (GMN)* we also evaluate our method on only predicting the masks, leaving the unmasked parameters intact. Additional implementation details are provided in appendix A.4.1.

As shown in figs. 1 and 6 in both datasets, we observe that our method, *GMN*, consistently dominates the Pareto front. The margin between *GMN* and the rest of the methods is larger when masking more features, which is expected since in those cases, stronger editing is needed. Moreover, the difference between *FS* and *FS (GMN)* is limited to the method used for computing the mask, since the un-masked parameters are not affected. We can see, however, that in some cases *FS (GMN)* performs better, which indicates that predicting a mask *per model* based on its parameters, finds the specific feature each model relies mostly on. In the edge cases, selecting a very small $\lambda \to 0$ results in no masking (no masked features) and zero function divergence, for the *GMN* case, leading the metanetwork-based methods to coincide with the baseline ones.

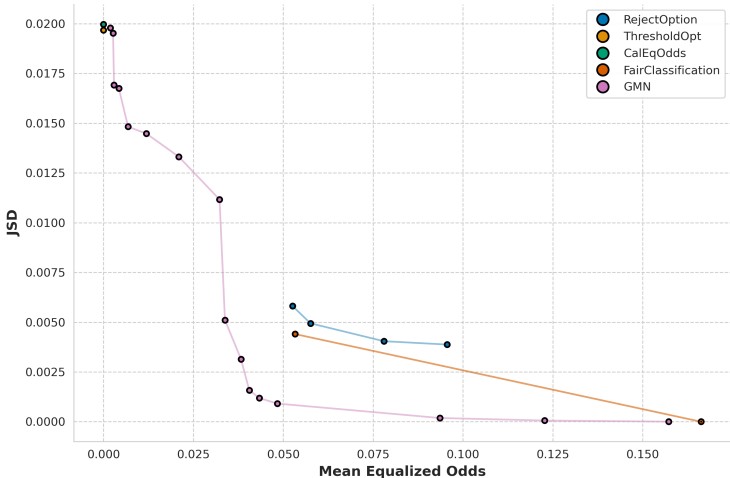

Figure 2: Bias Mitigation on the Adult UCI dataset. Sensitive attribute: gender

**Bias Mitigation.** Since our method poses a de facto post-processing method for bias mitigation, we consider as baselines methods that operate under the same regime. We select the traditional post-processing algorithms *ThresholdOpt* (Hardt et al., 2016) and *RejectOption* (Kamiran et al., 2012) and also *FairCls* (Xian et al., 2023). More details can be found in the appendix A.4.2. As shown in fig. 2, our method dominates across the whole Pareto front. In particular, it achieves lower JS Divergence for equivalent values of the EOD metric compared to the baseline, while also covering a wider area on the $x-$ axis. In particular, we were not able to achieve lower EOD using the *RejectOption* (Kamiran et al., 2012) and *FairCls* (Xian et al., 2023) methods. Moreover, *ThresholdOpt* (Hardt et al., 2016) and *CalEqOdds* (Pleiss et al., 2017) impose a hard equality constraint on equalized odds, resulting in a single point on the plot.

**Pruning.** Finally, we evaluate our method on pruning individual weights from trained models. We assess the performance of our metanetwork in two settings: pruning only (*GMN - Prune*) and pruning combined with editing (*GMN - Prune & Edit*). We compare our method with four baseline methods, each evaluated independently on the models of the test split of $\mathcal{D}_m$. *Random* prunes weights randomly, while *Grad Importance* prunes weights based on the gradient magnitude. Moreover, we compare our method with simpler versions (reducing the number of iterations) of *SNIP* (Lee et al., 2019) and *Lottery Ticket* (Frankle & Carbin, 2019). As shown in fig. 3, both of our methods dominate the Pareto front. As expected, the impact of editing the remaining parameters becomes particularly pronounced under high sparsity constraints, i.e. when the proportion of unmasked parameters is low.

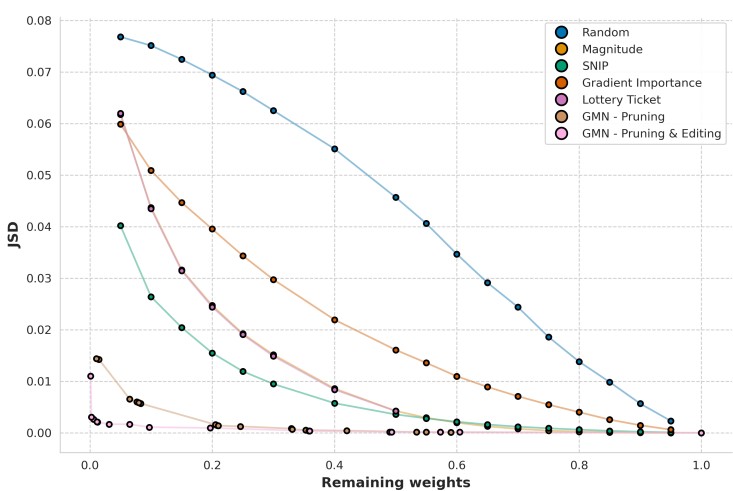

Figure 3: Pruning Adult UCI dataset

Table 1: Comparison of Methods and Computation Times

| Data Minimization Methods | | Bias Mitigation Methods | | Pruning Methods | |
|---|---|---|---|---|---|
| **Method** | **Time (s)** | **Method** | **Time (s)** | **Method** | **Time (s)** |
| *FS* | 32 | *ThresholdOpt* | 0.08 | *Random* | 0.003 |
| *FS & Retrain* | 32.35 | *CalEqOdds* | 0.37 | *Magnitude* | 0.003 |
| *FS (GMN)* | 0.03 | *RejectOption* | 4.36 | *Grad Importance* | 0.02 |
| *GMN* | **0.03** | *FairCls* | 0.17 | *SNIP* | 0.41 |
| | | *GMN* | **0.03** | *Lottery Ticket* | 0.05 |
| | | | | *GMN - Prune* | 0.03 |
| | | | | *GMN - Prune & Edit* | **0.03** |

**Time Efficiency Comparison.** Table 1 compares the time needed to edit an NN. As expected, our method is significantly faster than most baseline methods since it amounts to a single inference step of the metanetwork. It is slower only when compared to certain naive pruning techniques, which, as can be seen from the relevant figures, significantly compromise performance.

**Training Sample Efficiency** Although our framework offers fast model edits at inference, it depends on training a metanetwork on a population of trained models. To evaluate the training sample efficiency of our approach, we conducted an ablation study examining the impact of training set size on metanetwork performance. We focus on the pruning task as a representative case study.

We trained separate metanetworks using $10\%$, $25\%$, $50\%$, $75\%$, and $100\%$ of the training data from $\mathcal{D}_{\mathrm{m}}^{Adult}$. For each training set size, we maintained fixed hyperparameters and trained metanetworks for specific $\lambda$ values to construct the Pareto front. Critically, the test split remained identical across all experiments, ensuring fair comparison. All other experimental conditions (metanetwork architecture, optimization procedure, evaluation protocol) were kept constant.

The fig. 4 presents the Pareto fronts for different training set sizes alongside baseline pruning methods. The metanetwork exhibits strong sample efficiency across different training set sizes. Even with only $10\%$ of the training data, our method outperforms all baselines except at edge cases. At $25\%$ of the data, GMN substantially outperforms all baselines across the entire Pareto front. Notably, the gap between $50\%$, $75\%$, and $100\%$ training data is minimal, suggesting that the metanetwork achieves effective generalization with moderate dataset sizes. This indicates that approximately a quarter of the training data is sufficient to approach optimal performance.

**Limitations.** Training the metanetwork for requirement compliance necessitates a training dataset of NNs. This condition is more realistic when considering NNs trained for different tasks. Currently,

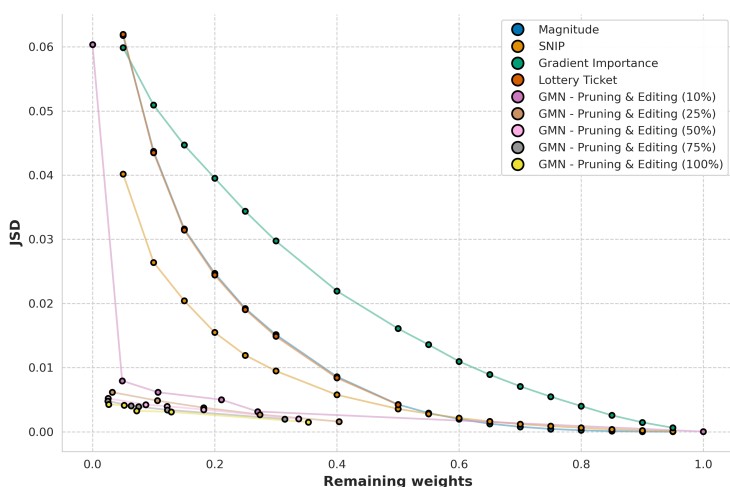

Figure 4: Ablation on the size of the training set

as discussed in section 4, our method is limited by the fact that it can deal with NN populations trained on a common task. Additionally, evaluation on complex architectures other than MLPs has been left for future work. Further technological advancements in the field of weight space learning are foreseen to provide auditors with the tools to address both the aforementioned challenges.

**Automated model editing & human oversight** In this work, we propose a mathematical framework for requirement compliance. Nevertheless, we emphasise that complete automation is beyond our intended scope and is not recommended, particularly in sociotechnical contexts. Although our framework generates interpretable Pareto fronts that enable systematic exploration of requirement-performance trade-offs, human judgment remains essential. Domain experts must evaluate edited models, assess trade-offs according to domain-specific priorities and constraints, and make final deployment decisions. This necessity for human oversight extends to automated compliance systems more broadly and represents a critical consideration for responsible AI deployment and governance.

## 6 CONCLUSION

In this work, we introduce a learnable NN editing paradigm as a unifying framework for requirement compliance. We demonstrate the versatility of our method through the lens of three different tasks, namely data minimisation, bias mitigation and weight pruning. With our work, we aspire to introduce a new research direction: leveraging weight space learning methods for automated neural network editing. We envision this paradigm as a foundation for future work on building adaptive post-processing tools that ensure trained models meet evolving regulatory, ethical, and performance requirements without necessitating costly retraining.

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

# A   APPENDIX

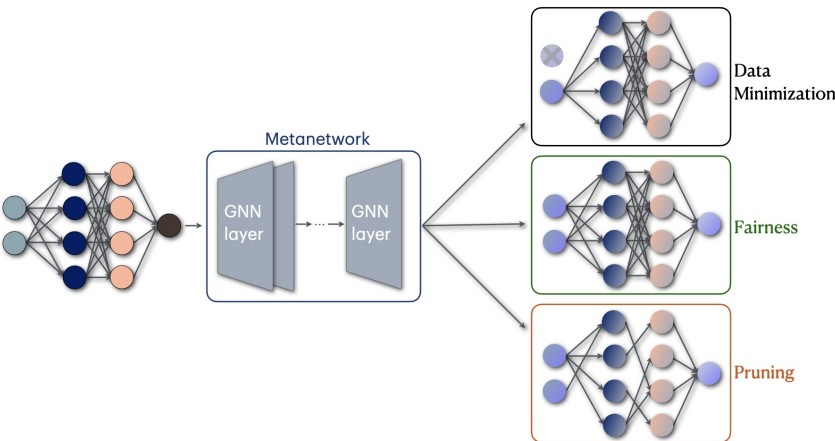

Figure 5: A unifying framework for learnable NN requirement compliance.

## A.1   REQUIREMENT OBJECTIVES

Our method can enforce different types of requirements by simply modifying the requirement objective within the training loss. Below, we provide a more detailed analysis of how each requirement is formulated and implemented in a differentiable manner within our framework.

### A.1.1   DATA MINIMISATION PRINCIPLE

The metanetwork is responsible for predicting a mask and simultaneously editing the model to compensate for the missing features. While the masking can be considered to be task-specific, we opt to predict the mask through the metanetwork and by accessing only the parameters of the input NN. We refrain from feeding the metanetwork with any task-related features, such as statistics or task representations (Jomaa et al., 2021), which could be easily computed using a DSS model (Maron et al., 2020). On the contrary, we first apply the metanetwork and subsequently perform node classification on the input nodes based on the computed features. After a sufficient number of GNN layers of the bidirectional variant from (Kalogeropoulos et al., 2024), the input nodes are expected to capture the information of the entire model. This approach predicts a mask for each model, rather than for each task.

From eq. (5), $\mathbf{m}$ is essentially the predicted mask on the input features and $\mu$ is simply responsible for applying this mask on the outgoing weights from the input nodes. Since masking constitutes part of the optimization process, directly applying hard masks undermines differentiability and hinders gradient flow. To address this limitation, we employ a relaxation to obtain $\mathbf{m} \in [0, 1]$.

Formally, consider $\mathbf{z}_i \in \mathbb{R}^k$ being the output of the node classifier $f_{\text{cls}}$ where $k = 2$ is the number of classes. For each element $i$ and each class $k \in \{0, 1\}$, we divide the logits by a temperature $\tau > 0$ controlling the discreteness, before applying the softmax. The soft sample is obtained via:

$$\mathbf{m}_i = \mathbf{y}_i^{\text{soft}} = \text{softmax}(\tilde{\mathbf{z}}_i) \qquad (15)$$

**Parameter symmetries.** Our training objective should be invariant to parameter symmetries. Here, we use a metanetwork equivariant to permutation symmetries; therefore, these will be the ones considered in the objective as well. Regarding this task, it is easy to see that since we are interested in the input node count, which remains unaffected by node permutations, the desideratum is satisfied.

### A.1.2 FAIRNESS

The multi-objective can now be written as:

$$\min_{\boldsymbol{\theta}'} \left( \mathbb{E}_{(\mathbf{x},y,s)\sim p_{\mathrm{d}}} \Big[ \mathrm{JSD}\Big( f_{G,\boldsymbol{\theta}}(\mathbf{x}), f_{G',\boldsymbol{\theta}'}(\mathbf{x}) \Big) \Big], \mathrm{EOD}(f_{G',\boldsymbol{\theta}'}, p_{\mathrm{d}}) \right), \tag{16}$$

When including the EOD in our objective, $\mathrm{TPR}_{i,k}$ and $\mathrm{TFR}_{i,k}, i \in \mathcal{S}, k \in \mathcal{K}$ depend on the predictions of the edited model. Specifically:

$$\hat{y}(\mathbf{x}) = \mathrm{argmax}_k(f_{G,\tilde{\boldsymbol{\theta}}}(\mathbf{x})) \tag{17}$$

To acquire model predictions without hindering the differentiability of our method, we apply a softmax-with-temperature relaxation to obtain the soft predictions:

$$\hat{y}_{\mathrm{soft}}(\mathbf{x}) = \mu(f_{G,\tilde{\boldsymbol{\theta}}}(\mathbf{x})) = \mathrm{softmax}\left( \frac{f_{G,\tilde{\boldsymbol{\theta}}}(\mathbf{x})}{\tau} \right), \tag{18}$$

where $\tau > 0$ the temperature parameter.

**Parameter symmetries.** Observe that the requirement objective depends only on the outputs of the model. As a result, it inherits its symmetries, i.e. it remains unaffected when applying any valid parameter symmetry, including permutations.

### A.1.3 PRUNING

Similarly to Data Minimisation, the metanetwork is responsible for predicting the pruning of individual weights and for editing the remaining parameters to limit the functional deviation. In this scenario, we apply edge classification on the updated edge representations. By construction, the weights of the MLP coincide with the edges of the constructed parameter graph; hence, this approach essentially predicts which edges/weights should be pruned. Again, since predicting hard masks hinders the gradient flow, we resolve the differentiability issue using softmax-with-temperature $\tau > 0$.

**Parameter symmetries.** Similar to the Data Minimisation case, the requirement objective contains only the number of edges, which remains unaffected by permutation symmetries. Therefore, the invariance desideratum is satisfied here as well.

### A.2 METANETWORKS

**Metanetwork preliminaries.** In this paper, we focus our analysis on Feedforward Neural Networks (FFNNs), i.e. linear layers interleaved with non-linearities. Consider NNs of the form $f_{G,\boldsymbol{\theta}} : \mathcal{X} \to \hat{\mathcal{X}}$, where $\mathcal{X} = \mathbb{R}^{d_{\mathrm{in}}}$ and $\hat{\mathcal{X}} = \mathbb{R}^{d_{\mathrm{out}}}$ of the following form:

$$\mathbf{x}_0 = \mathbf{x}, \quad \mathbf{x}_\ell = \sigma_\ell\left( \mathbf{W}_\ell \mathbf{x}_{\ell-1} + \mathbf{b}_\ell \right), \quad f_{G,\boldsymbol{\theta}}(\mathbf{x}) = \mathbf{x}_L \tag{19}$$

where $L$: the number of layers, $\mathbf{W}_i \in \mathbb{R}^{d_\ell \times d_{\ell-1}}$: the weights of the NN, $\mathbf{b}_i \in \mathbb{R}^{d_\ell}$: the biases of the NN, $d_0 = d_{\mathrm{in}}, d_L = d_{\mathrm{out}}, \sigma_\ell : \mathbb{R} \to \mathbb{R}$ activation functions applied element-wise. Here, the learnable parameters are $\boldsymbol{\theta} = (\mathbf{W}_1, \ldots, \mathbf{W}_L, \mathbf{b}_1, \ldots, \mathbf{b}_L)$ and the computational graph encodes the connections between vertices and the type of activations used in each layer.

The connectivity of the computational graph, together with the choice of the activation functions, induces parameter symmetries, an area that has been extensively, long before the advent of metanetworks. Indicatively, the interested reader can refer to the works of Hecht-Nielsen (1990); Chen et al. (1993); Godfrey et al. (2022) to study the permutation and scaling symmetries that arise.

**Metanetwork Architecture:** To train on a dataset of heterogeneous architectures, we employ Graph Metanetworks as proposed by Lim et al. (2024), Kofinas et al. (2024), and Kalogeropoulos et al.

(2024). In particular, our approach adapts the bidirectional variant from Kalogeropoulos et al. (2024), focusing, however, on the permutation symmetries only. Since we are interested in an editing task, the bidirectional nature of the model is crucial to propagate information across the whole graph. To handle the more complex MLP architectures, we adapt the Graph Constructor. For more details, please refer to the appendix A.3.1. Finally, as it is usually easier, we use the metanetwork to predict the residuals $\Delta\boldsymbol{\theta}$ and the new parameters are given as $\hat{\boldsymbol{\theta}} = \boldsymbol{\theta} + \gamma\mathcal{F}(G, \boldsymbol{\theta}; \boldsymbol{\phi})$, where $\gamma$ is also a learnable parameter.

## A.3 IMLPEMENTATION DETAILS

### A.3.1 GRAPH CONSTRUCTION

We closely follow the dataset construction and initialization procedure from Kalogeropoulos et al. (2024). In the case, of MLPs, the parameter graph coincides with the computational graph of the model. In particular, let $G = (\mathcal{V}, \mathcal{E})$ be the computational graph, $i \in \mathcal{V}$ an arbitrary vertex in the graph (neuron) and $(i, j) \in \mathcal{E}$ an arbitrary edge from vertex $j$ to vertex $i$. The biases are mapped to the vertex features, denoted as $\mathbf{x}_V \in \mathbb{R}^{|\mathcal{V}| \times d_v}$, while the weights are assigned to the edge features $\mathbf{x}_E \in \mathbb{R}^{|\mathcal{E}| \times d_e}$.

**Positional Encodings** We apply two types of positional encodings. The first one accounts for the fact that not all permutations are valid. In particular, only vertices corresponding to hidden neurons are permutable within the same hidden layer. Likewise, edges of hidden layers can only be permuted within the same layer. In-coming (out-going) edges to (from) an input or output neuron can only be permuted with in-coming (out-going) edges to (from) the same neuron. Hence, we apply symmetry-breaking positional encodings $(\mathbf{p}_V^s, \mathbf{p}_E^s)$ to account for the above behavior, similarly to Kalogeropoulos et al. (2024). Moreover, since our dataset consists of MLPs of varying architectures and hyperparameters, see appendix A.6, distinguishing between two models of the same structure (connectivity) but with different modules (activation functions or normalization modules), is a necessary property of our metanetwork. For this reason, we employ positional encodings to account for the functionality of each node and edge $(\mathbf{p}_V^f, \mathbf{p}_E^f)$, similarly to Lim et al. (2024). Finally, the initialization of the vertex and edge representations is shown below:

$$\mathbf{h}_V^0(i) = \text{INIT}_V\left(\mathbf{x}_V(i), \mathbf{p}_V^s(i), \mathbf{p}_V^f(i)\right), \quad \mathbf{h}_E^0(i) = \text{INIT}_E\left(\mathbf{x}_E(i, j), \mathbf{p}_E^s(i, j), \mathbf{p}_E^f(i, j)\right), \tag{20}$$

where INIT is a general function approximator (e.g. MLPs).

### A.3.2 COMPARING ON THE FUNCTION SPACE

Functionality preservation necessitates the need to compare the original $f_{G,\boldsymbol{\theta}}$ and edited $f_{\hat{G},\hat{\boldsymbol{\theta}}}$ models on their function space, while accessing the function space is also needed for the requirements of Bias Mitigation. Evaluating the model's outputs on the whole split is prohibitively time-consuming. Hence, on each step, we simply sample $k$ data points.

### A.3.3 STRAIGHT-THROUGH ESTIMATOR

To preserve gradient flow while using discrete masks, we employ the straight-through estimator. During the forward pass, we use the hard sample, while during the backward pass, gradients flow through the soft sample:

$$\text{Forward:} \quad \mathbf{m}_i = \mathbf{y}_i^{\text{hard}} \tag{21}$$

$$\text{Backward:} \quad \frac{\partial\mathcal{L}}{\partial\mathbf{z}_i} = \frac{\partial\mathcal{L}}{\partial\mathbf{m}_i}\frac{\partial\mathbf{y}_i^{\text{soft}}}{\partial\mathbf{z}_i} \tag{22}$$

This is implemented as:

$$\mathbf{m}_i = \mathbf{y}_i^{\text{hard}} - \text{sg}(\mathbf{y}_i^{\text{soft}}) + \mathbf{y}_i^{\text{soft}} \tag{23}$$

where $\text{sg}(\cdot)$ denotes the stop-gradient operation.

## A.4 EXPERIMENTAL DETAILS

As described in section 5, we maintain train, validation and test splits for both $\mathcal{D}_\text{m}$ and $\mathcal{D}_\text{d}$ on each task. For training our metanetwork, we use the train split of $\mathcal{D}_\text{m}$, using, however, a subset of the validation set of $\mathcal{D}_\text{d}$ to access the function space of the models. *This aims to use data samples unseen during the training of the original model.* The rest of the $\mathcal{D}_\text{d}$ validation split is used as usual during the evaluation. Finally, the test splits of both $\mathcal{D}_\text{m}$ and $\mathcal{D}_\text{d}$ are reserved only for testing.

Regarding our baselines, we evaluate them only on the test split, independently, on each data point of the split, as there is no learning across models in any of them. In the cases where data samples are needed during training of the baselines, we use the same split of the validation split of $\mathcal{D}_\text{d}$.

**Hyperparameters:** Our model is a GNN model on a *bidirectional* graph, as described by Kalogeropoulos et al. (2024), while we also use the official implementation by the authors in Py-Torch (Paszke et al., 2019). In all of the experiments, we search over the following hyperparameters: batch size in $\{32, 64\}$, hidden dimension in $\{64, 128\}$, learning rate in $\{5e-5, 1e-3\}$, dropout in $\{0.0, 0.1, 0.2\}$, weight decay in $\{0.0, 1e-5, 1e-4\}$ and number of GNN layers (depth) in $\{5, 6\}$. The learnable parameter $\gamma$ was initialized at $0.1$. Finally, for $\lambda$ we searched in $[1e-4, 2]$.

### A.4.1 DATA MINIMIZATION

In our evaluations, we used two baselines, namely *FS* and *FS & Retrain*. As a first step, both of them apply Permutation Feature Importance (PFI) Breiman (2001) to sort input features by their relevance for each dataset. For the *FS* baseline, we directly feed the masked input to the original model. For the *FS & Retrain* baseline, we use a knowledge distillation-based approach Hinton et al. (2015). In particular, we assign the original model as the teacher, training a student model that receives only masked inputs. Instead of the traditional weighted loss, the training objective uses Jensen-Shannon (JS) divergence (Lin, 2002) as the loss function. We train the student for 100 epochs. Works that approached the Data Minimization Principle, from another however perspective, are (Goldsteen et al., 2022; Tran & Fioretto, 2023; Ganesh et al., 2025).

### A.4.2 BIAS MITIGATION

For our experiments, we used the implementation from the FairLearn library (Bird et al., 2020) for the *ThresholdOpt* baseline and the library (Bellamy et al., 2018) for the *RejectOption* and *CalEqOdds* baselines. Finally, for *FairCls* we used the official implementation from Xian et al. (2023). For all the baselines, we result in different points in the Pareto by sweeping over hyperparameters and the fairness tolerance of each method.

All the aforementioned frameworks were implemented in TensorFlow, hence we had to integrate the above methods to our PyTorch implementation.

## A.5 EXPERIMENTS ON BANK DATASET

**Data Minimization**

Similarly to the Adult dataset, we observe that our method significantly outperforms all the baselines, while the baselines follow the same trends.

**Pruning**

In line with the observations on the Adult dataset, we see that both *GMN - Prune* and *GMN - Prune & Edit* outperform the baselines.

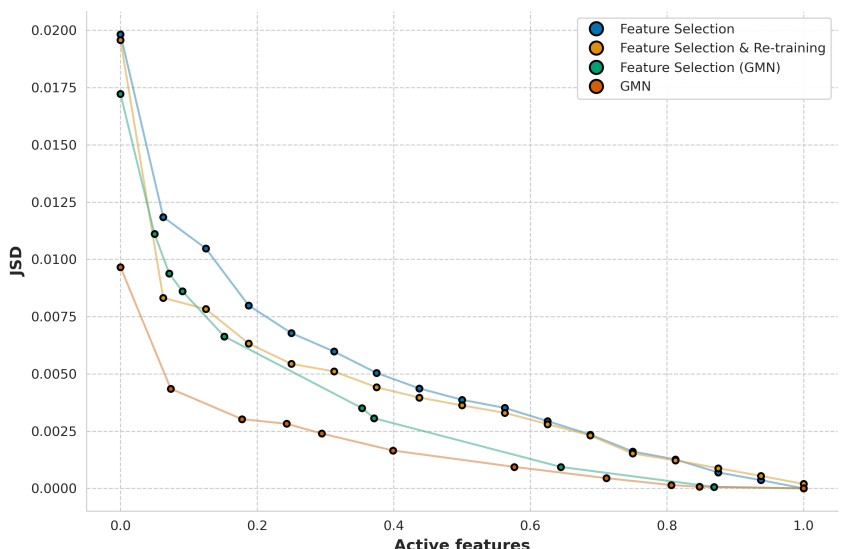

Figure 6: Data Minimization on $\mathcal{D}_{\mathrm{m}}^{Bank}$.

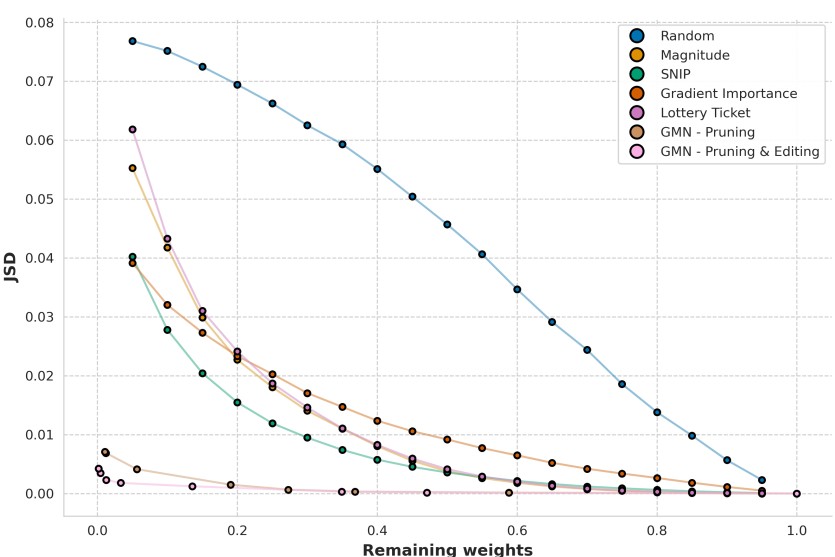

Figure 7: Weight pruning on $\mathcal{D}_{\mathrm{m}}^{Bank}$.

### A.6 DATASET CONSTRUCTION

We use the datasets *Adult* and *Bank Marketing* from the UCI repository (Kelly et al., 2024). This repository contains tabular datasets from various domains. (Fernández-Delgado et al., 2014) provided a pre-processed by version of 121 of these tasks, publicly available at [3]. We opted to use this pre-processed version, as it provised a common method of standardizing the datasets, keeping however, the number of features intact and not using techniques such as one-hot-encoding.

**Sampling Strategy.** For each task of the UCI dataset repository, we sample configurations, consisted of various hyperparameters and architecture designs, and train the resulting model on the given task. The list of hyperparameters used are listed in table 2. To sample a complete experiment, we first sample the number of layers of the MLP. Then, for each layer, we sample its hidden size, ensuring,

---
[3] http://www.bioinf.jku.at/people/klambauer/data_py.zip

however, that the hidden size of the model increases monotonically. That means we exclude from the sampling choices the sizes that are smaller than the current hidden size. Subsequently, we sample the hyperparameters that are global to the whole model, such as dropout, activation function, and normalization method. Then, for each hidden layer, we sample the number of the incoming skip connections and the source of each connection, strictly from one of the previous layers. The final layer is always of size $d_{out}$, where $d_{out} = \#$classes. That means that even for binary classification tasks, the sampled network is a function $f : \mathbb{R}^{d_{in}} \rightarrow \mathbb{R}^{d_{out}}$ and is trained using Cross Entropy Loss. Finally, we sample the training-related hyperparameters, namely learning rate and weight decay.

Table 2: Hyperparameters and their sampling strategies.

| Name | Distribution | Range |
|------|--------------|-------|
| out features | fixed | # classes |
| depth | choice | $\{1, 2, 3, 4\}$ |
| hidden dimension | choice | $\{32, 48, 64\}$ |
| dropout | choice | $\{0., 0.1, 0.2, 0.3\}$ |
| final activation function | fixed | identity |
| normalization | choice | $\{$BatchNorm1d, LayerNorm, null$\}$ |
| bias | fixed | True |
| skip connections | choice | $\{0, 1\}$ |
| activation function | choice | $\{$relu, gelu, tanh, sigmoid, leaky_relu, identity$\}$ |
| batch size | choice | $\{64, 128, 256\}$ |
| learning rate | log uniform | $[1e-3, 1e-1]$ |
| weight decay | log uniform | $[1e-6, 1e-2]$ |
| seed | random | - |

For each sampled configuration, we train for 60 epochs. In every experiment, we save three checkpoints: one early, one at the midpoint, and one at the final epoch. We sample 4000 experiments in total, which results in 12000 data point models for each of the two datasets *Adult* and *Bank Marketing*.

A.7   COMPOSING METANETWORKS.

Having validated training metanetworks on various objectives, an intriguing question arises: *Can we stack trained metanetworks to edit towards more than one requirement?* Formally, this can be defined as $\mathfrak{F} = \mathfrak{F}_N \circ \cdots \mathfrak{F}_1$, where each $\mathfrak{F}_n : \mathcal{G} \times \Theta \rightarrow \mathcal{G} \times \Theta$, for $n \in \{1, \cdots, N\}$ represents a metanetwork trained on a distinct objective. A natural assumption, however, of stacking metanetworks is that each $\mathfrak{F}_n$ does not drastically alter the distribution of the input model parameters. Specifically, we assume that the output $(G'_n, \boldsymbol{\theta}'_n) = \mathfrak{F}_n(G, \boldsymbol{\theta})$ of the n-th metanetwork remain within the original parameter distribution $p_m$, i.e., $(G'_n, \boldsymbol{\theta}'_n) \sim p_m$.

We evaluate the composition of a metanetwork $\mathfrak{F}_{dm}$ trained on Data Minimization with a metanetwork $\mathfrak{F}_{pr}$ trained on Pruning, to obtain $\mathfrak{F} = \mathfrak{F}_{pr} \circ \mathfrak{F}_{dm}$. We use a set of metanetworks trained on various $\lambda$ values for each objective, $\Lambda_{dm}$ and $\Lambda_{pr}$ respectively, and evaluate the composition of $\Lambda_{dm} \times \Lambda_{pr}$ metanetworks. In both cases, we *edit the remaining parameters*, as done in *GMN* and *GMN - Prune & Edit* respectively. For reference, we compare with the composition of the best baselines from the Data Minimization (*FS & Retrain*) and Pruning (*SNIP*) experiments. In fig. 8, we observe that our method fully dominates the Pareto fronts that occur, even when compared to the best alternatives. The outcome of this experiment functions as an additional empirical evaluation to our framework*: the fact that metanetworks trained independently can be composed effectively suggests that each metanetwork preserves the parameter distribution sufficiently well that subsequent metanetworks can operate on their outputs. In other words, this is empirical evidence that our preservation objective (minimizing JSD) preserves the NNs' functionality without causing a harmful side-effect (distribution shift).

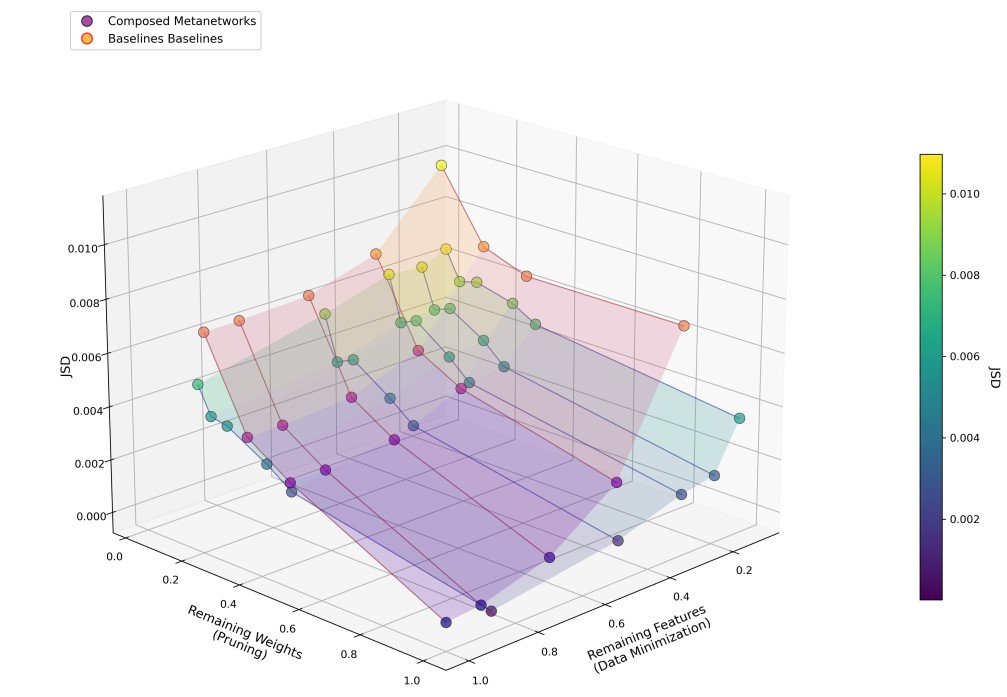

Figure 8: Composing metanetworks trained on different objectives, as $\mathfrak{F} = \mathfrak{F}_{pr} \circ \mathfrak{F}_{dm}$. Each line corresponds to the composition of a $\mathfrak{F}_{dm}$ trained using a $\lambda_{dm} \in \Lambda_{dm}$ with a metanetwork $\mathfrak{F}_{pr}$ on various lambda values $\lambda_{pr} \in \Lambda_{pr}$. 30 compositions in total, with $|\Lambda_{dm}| = 5$ and $|\Lambda_{pr}| = 6$.

## A.8 EPO vs LS

Linear Scalarization has a fundamental limitation: it cannot find solutions in concave regions of the Pareto front, leading to suboptimal solutions in non-convex multi-objective problems. To address this, we also employ the Exact Pareto Optimal (EPO) Search Mahapatra & Rajan (2020) algorithm, which makes no convexity assumptions and can identify exact solutions for any preference vector.

In fig. 9, we observe that *GMN - EPO* yields a Pareto front qualitatively similar to that obtained with *GMN - LS*. Moreover, *GMN - EPO* covered a substantial portion of the Pareto front with relatively few preference vectors, unlike LS, which requires more extensive exploration. Notably, our experiments evaluate model generalization to unseen data points, not just Pareto coverage. Consequently, while EPO guarantees exact Pareto optimal solutions during training, this optimality is not guaranteed on validation and test splits.

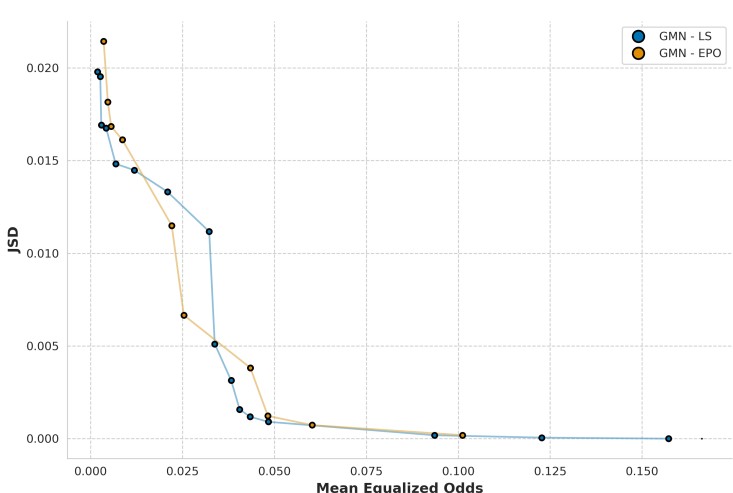

Figure 9: Comparing Linear Scalarization to Exact Pareto Optimal (EPO) Search Mahapatra & Rajan (2020).

