# OpenReview forum: "Metanetworks as Regulatory Operators: Learning to Edit for Requirement Compliance"
_ICLR.cc/2026/Conference — Submitted to ICLR 2026_

### Official Review · Reviewer_eYmo · 2025-10-25

**Soundness:** 3
**Presentation:** 3
**Contribution:** 2
**Rating:** 6
**Confidence:** 2

**Summary:**

Paper proposes a graph metanetwork that edits trained MLPs in one forward pass to satisfy requirements (data minimization, equalized-odds fairness, pruning) while preserving behavior. It outputs masks and residuals, trained over model families with JSD + requirement losses, yielding faster, better Pareto trade-offs on Adult/Bank.

**Strengths:**

1. Motivation is well-argued and the paper is clearly written, with a crisp problem setup and method description.

2. Covers three compliance needs—data minimization, equalized-odds fairness, and pruning—with promising Pareto trade-offs and extremely fast inference (~0.03s per edit).

3. Accommodates different parameter shapes by operating on a parameter graph, rather than assuming a fixed MLP architecture.

**Weaknesses:**

1. For each dataset, each requirement, and each weighting coefficient, a separate model must be designed and trained.

2. Given the data-hungry training (needing many trainings to construct weight samples) and poor cross-dataset generalization, the claim of “no costly retraining” seems hard to stand.

3. The datasets are extremely limited and very low-dimensional, which makes me worry about applicability to higher-dimensional real-world settings.

**Questions:**

I’d like to know how it performs on higher-dimensional datasets, and how much the final results depend on the number of pre-trained weight samples (weight snapshots) used for training？

---

> ### Author Response · Authors · 2025-11-26
> **Rebuttal by Authors**
>
> > Weakness 1: For each dataset, each requirement, and each weighting coefficient, a separate model must be designed and trained.
>
> We thank the reviewer for pointing this out. We examine each case separately:
>
> **Requirement**: As detailed in Section 3.2, the metanetwork's objective function is explicitly tailored to each specific requirement. Since each requirement constitutes a distinct optimization problem with different objective formulations, training separate metanetworks from scratch for each requirement is inevitable.
>
> **Dataset**: As also discussed in Reviewer's 4ehz Question 3 and also detailed in the Remark of Section 4 and in the limitations of our work in the manuscript, our metanetworks, in their current form, are not designed to generalize across different tasks. Achieving this level of generalization *presents technical challenges that the current weight-space learning literature has not yet tackled*. However, we envision that addressing them, will provide us with the necessary toolbox to extend our work to simultaneously learning from multiple tasks.
>
>
> **Weighting Coefficient**: A promising direction for future work involves conditioning the metanetwork on the $\lambda$ value, yielding a function of the form $\mathfrak{F}(\mathcal{G}, \pmb{\theta}, \lambda)$. By sampling weighting coefficients during training, the metanetwork could generalize across $\lambda$ values, thereby eliminating the need to train separate models for each point on the trade-off curve. This approach, commonly referred to as "Learning the Pareto front" [1] in the multi-objective optimization literature, represents a natural and attractive extension of our framework that we leave to future work.
>
>
>
> > Weakness 2: Given the data-hungry training (needing many trainings to construct weight samples) and poor cross-dataset generalization, the claim of “no costly retraining” seems hard to stand.
>
> We thank the reviewer for pointing out the over-head of training a metanetwork. Regarding the dataset construction, a fundamental motivation behind weight space learning is the exploitation of already-trained, publicly-available model repositories (e.g. Hugging Face). Generalizing across different tasks, as discussed in Weakness 1, will substantially simplify dataset creation and, in turn, facilitate training a single metanetwork on models originating from diverse task distributions.
>
> We emphasize that this work represents an initial step toward learning to perform requirement-compliant model edits. We anticipate that continued progress in weight-space learning will equip auditors with increasingly effective tools to address the challenges outlined above.
>
> Finally, we note that the primary efficiency gains of our framework manifest at inference time: once trained, the metanetwork executes edits significantly faster than existing alternatives.
>
> > Weakness 3:     The datasets are extremely limited and very low-dimensional, which makes me worry about applicability to higher-dimensional real-world settings.
>
> We acknowledge this limitation and appreciate the reviewer raising this point. Our choice to focus on tabular data and MLPs was deliberate and grounded in two key considerations:
> 1. **Compatibility with all studied requirements**: Tabular data provides a natural testbed where Data Minimization and Fairness can be directly evaluated alongside Pruning. This allowed us to demonstrate the versatility of our framework across diverse regulatory requirements.
> 2. **Current scope of metanetwork applications**: From the weight space learning perspective, metanetworks have only been successfully employed in limited editing scenarios to date—primarily INR Editing [2,3,4] and Learned Optimization [3,4,5]. Applying metanetworks to the challenging task of requirement compliance represents a significant conceptual advance, and we opted to establish this proof-of-concept on MLPs before tackling more complex architectures.
>
> Moreover, the proposed framework itself is architecture-agnostic, by injectively mapping NNs to graphs and processing them using GNNs. Recent works [3,6] have demonstrated this capability for various architectures, providing a clear path for extension.
> Nevertheless, we are actively working on extending our pruning experiments to CNNs and plan to include these results in a revised manuscript.

---

> > ### Author Response · Authors · 2025-11-26
> > **Rebuttal by Authors**
> >
> > > Question 1: Performance on higher-dimensional datasets
> >
> > Please refer to response to Weakness 3
> >
> > > Question 2: How much the final results depend on the number of pre-trained weight samples (weight snapshots) used for training
> >
> > To examine the data requirements of our framework, we conduct an ablation study focusing on the pruning task. We trained separate metanetworks using 10%, 25%, 50%, 75%, and 100% of the training data from $D_m^{Adult}$ and evaluated them on the original test set. The original size of the training set is 12000 models.
> >
> > The results shown in Figure 4 in section 5 of the updated manuscript present the Pareto fronts for different training set sizes alongside baseline pruning methods. The metanetwork exhibits strong sample efficiency across different training set sizes. **Even with only 10% of the training data, our method outperforms all baselines** except at edge cases. At 25% of the data, GMN substantially outperforms all baselines across the entire Pareto front. Notably, the gap between 50%, 75%, and 100% training data is minimal, suggesting that the metanetwork achieves effective generalization with moderate dataset sizes. This indicates that approximately a **quarter (~3000 models) of the training data is sufficient** to approach optimal performance.
> >
> >
> > Further details regarding the ablation can be found in the section 5 of the revised manuscript.
> >
> > ---
> >
> > [1] Navon, Aviv, et al. "Learning the Pareto Front with Hypernetworks." International Conference on Learning Representations.
> >
> > [2] Zhou, Allan, et al. "Permutation equivariant neural functionals." Advances in neural information processing systems 36 (2023): 24966-24992.
> >
> > [3] Kofinas, Miltiadis, et al. "Graph Neural Networks for Learning Equivariant Representations of Neural Networks." The Twelfth International Conference on Learning Representations.
> >
> > [4] Gelberg, Yoav, et al. "GradMetaNet: An Equivariant Architecture for Learning on Gradients." arXiv preprint arXiv:2507.01649 (2025).
> >
> > [5] Zhou, Allan, Chelsea Finn, and James Harrison. "Universal neural functionals." Advances in neural information processing systems 37 (2024): 104754-104775.
> >
> > [6] Lim, Derek, et al. "Graph Metanetworks for Processing Diverse Neural Architectures." The Twelfth International Conference on Learning Representations.

---

> > > ### Comment · Reviewer_eYmo · 2025-11-26
> > >
> > > Thank you for the response. It addresses my concerns to a large extent. Although my worries about generalization ability and data dimensionality are not completely resolved, I acknowledge the challenges involved and agree that expecting all of these issues to be fully settled in a single work would be overly demanding. I believe this paper is a good first step in this direction. I have therefore increased my confidence score to 3.

---

### Official Review · Reviewer_4ehz · 2025-10-31

**Soundness:** 3
**Presentation:** 3
**Contribution:** 3
**Rating:** 4
**Confidence:** 3

**Summary:**

This paper investigates the problem of ensuring ML models comply with evolving requirements such as fairness, privacy, and efficiency without retraining or performance degradation. It identifies that existing post-processing and fine-tuning approaches are inefficient or limited to specific applications, highlighting a gap in flexible, general-purpose model editing.
To address this, the authors propose a graph metanetwork that learns to edit other neural networks’ parameters directly, performing requirement-compliant modifications in a single inference step. The framework formulates requirement compliance as a multi-objective optimization problem balancing performance preservation and regulatory constraints, trained in a data-driven and symmetry-aware manner. Experiments across data minimization, bias mitigation, and pruning show the method achieves better trade-offs between performance, compliance, and computational efficiency than retraining or post-processing baselines.

**Strengths:**

1.	The motivation is strong and well-timed, the paper tackles a pressing issue in trustworthy AI, how to make models meet regulatory and ethical requirements post-deployment.
2.	The metanetwork architecture is innovative: using an equivariant GNN to operate in weight space is a cutting-edge idea that leverages recent advances in meta-learning and symmetry-preserving design.
3.	Experiments are thoughtfully structured across diverse requirements (fairness, data minimization, pruning), which convincingly demonstrate versatility rather than one task.

**Weaknesses:**

1.	The scope of experiments is limited to MLPs and tabular data. This makes it unclear how well the method scales to large models (e.g., Transformers, diffusion, or other large models) or structured modalities such as text and vision.
2.	The assumption of white-box access to model weights may restrict applicability in auditing or proprietary systems, where only API access is available.
3. The training cost of the metanetwork itself, though amortized at inference, is not deeply analyzed. It remains uncertain how expensive or data-hungry the pre-training process is compared to fine-tuning.
4. It would be beneficial if further interpretability analysis and error analysis could be involved.

**Questions:**

1. The paper mainly focuses on differentiable objectives, how to consider other requirement types like safety constraints, robustness?
2. How realistic is this in common auditing or deployment settings, where many models are only available via APIs?
3. Can the metanetwork generalize across tasks?
4. How sensitive is performance to the diversity and size of this NN population?

---

> ### Author Response · Authors · 2025-11-26
> **Rebuttal by Authors**
>
> > Weakness 1: The scope of experiments is limited to MLPs and tabular data.
>
> We acknowledge this limitation and appreciate the reviewer raising this point. Our choice to focus on tabular data and MLPs was deliberate and grounded in two key considerations:
> 1. **Compatibility with all studied requirements**: Tabular data provides a natural testbed where Data Minimization and Fairness can be directly evaluated alongside Pruning. This allowed us to demonstrate the versatility of our framework across diverse regulatory requirements.
> 2. **Current scope of metanetwork applications**: From the weight space learning perspective, metanetworks have only been successfully employed in limited editing scenarios to date—primarily INR Editing [1,2,3] and Learned Optimization [2,3,4]. Applying metanetworks to the challenging task of requirement compliance represents a significant conceptual advance, and we opted to establish this proof-of-concept on MLPs before tackling more complex architectures.
>
> Moreover, the proposed framework itself is architecture-agnostic, by injectively mapping NNs to graphs and processing them using GNNs. Recent works [2,5] have demonstrated this capability for various architectures, providing a clear path for extension.
> Nevertheless, we are actively working on extending our pruning experiments to CNNs and plan to include these results in a revised manuscript.
>
>
> > Weakness 2: The assumption of white-box access to model weights may restrict applicability in auditing or proprietary systems, where only API access is available.
>
> We thank the reviewer for drawing the attention to this. We acknowledge that white-box access to model parameters may not always be available, particularly in external auditing scenarios involving proprietary systems. However, we emphasize several key points:
>
> First, white-box and black-box methods serve complementary roles and should work synergistically within a comprehensive auditing ecosystem, depending on the access constraints and auditing objectives. Second, there exist numerous scenarios where white-box access is granted, e.g. internal auditing procedures or regulatory frameworks that mandate model transparency. In such cases, parameter-editing methods provide efficient and precise *interventions that black-box approaches cannot achieve*. Third, as demonstrated by [6], black-box approaches alone are fundamentally insufficient for rigorous AI auditing, underscoring the necessity of white-box methods, like ours. In summary, although white-box access may not be universally available, research in this direction is essential for developing the comprehensive toolset needed for effective AI auditing and compliance.
>
> > Weakness 3: The training cost of the metanetwork itself, though amortized at inference, is not deeply analyzed. It remains uncertain how expensive or data-hungry the pre-training process is compared to fine-tuning.
>
> We thank the reviewer for pointing towards this direction. To examine the data requirements of our framework, we conduct an ablation study focusing on the pruning task as a representative case study. Specifically, we trained separate metanetworks using 10%, 25%, 50%, 75%, and 100% of the training data from $D_m^{Adult}$ and evaluated them on the original test set. As mentioned in Section A.6 of the appendix, the original size of the training set is 12000 models.
>
> The results shown in Figure 4 of the updated manuscript present the Pareto fronts for different training set sizes alongside baseline pruning methods. **Even with only 10% of the training data, our method outperforms all baselines** except at edge cases. At 25% of the data, GMN substantially outperforms all baselines across the entire Pareto front. Notably, the gap between 50%, 75%, and 100% training data is minimal, suggesting that the metanetwork achieves effective generalization with moderate dataset sizes. This indicates that approximately a **quarter of the training data** is sufficient to approach optimal performance.
>
>
> These findings demonstrate that while our approach requires an initial dataset of trained models, the data requirements are more modest than one might initially expect. To summarize, *the metanetwork's strong generalization from limited data, combined with its ability to edit unlimited models at inference time without retraining, demonstrates the practical viability of our framework.*
>
> Further details regarding the ablation can be found in section 5 of the revised manuscript.
>
> Again, we thank the reviewer for motivating us to examin further the data requirements of our method.

---

> > ### Author Response · Authors · 2025-11-26
> > **Rebuttal by Authors**
> >
> > > Weakness 4: It would be beneficial if further interpretability analysis and error analysis could be involved.
> >
> > Thank you for this suggestion. To further evaluate our framework we have included two ablation studies:
> >
> > 1. **Training Sample Efficiency** as described above
> > 2. **Composing metanetworks** trained on different requirement objectives (Section A.7 in the appendix of the updated manuscript.)
> >
> > Beyond the above ablation studies, could you specify what particular interpretability or error analysis you would find most beneficial? We would be pleased to provide additional targeted analysis in our rebuttal.
> >
> >
> > > Question 1: The paper mainly focuses on differentiable objectives, how to consider other requirement types like safety constraints, robustness?
> >
> > We thank the reviewer for this question, which allows us to clarify an important aspect of our work. While the metanetwork relies on gradient-based optimization and thus requires differentiability, the requirements we address are inherently *non-differentiable objectives*, as detailed in section 3.2. Specifically, for Data Minimization and Pruning, the prediction of hard masks, applied to nodes and edges respectively, introduces non-differentiability. Similarly, the Fairness objective depends on hard predicted labels from the edited model, which again hinders the differentiability. To overcome this challenge, we introduce continuous relaxations that enable gradient-based optimization. Moreover, we acknowledge that alternative approaches, such as reinforcement learning, could also be viable solutions worth exploring.
> >
> >
> >
> > > Question 2: How realistic is this in common auditing or deployment settings, where many models are only available via APIs?
> >
> > Please see the response to Weakness 2.
> >
> > > Question 3: Can the metanetwork generalize across tasks?
> >
> > We thank the reviewer for pointing to this important research direction. As discussed in the Remark of Section 4, enabling metanetworks to generalize across diverse tasks would significantly enhance the practical applicability of our framework, as it would allow a single metanetwork to enforce compliance requirements across models trained on different tasks. Moreover, dataset collection for training such a metanetwork would be substantially more feasible, as large repositories of models trained on diverse tasks already exist (e.g., on platforms like Hugging Face).
> >
> > However, achieving this level of generalization *presents substantial technical challenges that the current weight-space learning literature has not yet adequately addressed*. Such a metanetwork would need to process not only model parameters but also task-specific input-output pairs, while handling far greater heterogeneity in the neural network distribution. We view this as an important avenue for future work that requires careful methodological development before it can be rigorously evaluated.
> >
> > > Question 4: How sensitive is performance to the diversity and size of this NN population?
> >
> > As discussed in the Section A.6 of the Appendix, we construct a diverse dataset of MLPs, by sampling architectures of varying hyperparameters. The empirical evaluation demonstrates that the Graph Metanetwork is able to generalize in all the requirements that we examined, while the newly added ablation study on the training size, corroborates our claim that learnable editing can be done efficiently, with minimal performance sacrifices.
> >
> > ---
> >
> > [1] Zhou, Allan, et al. "Permutation equivariant neural functionals." Advances in neural information processing systems 36 (2023): 24966-24992.
> >
> > [2] Kofinas, Miltiadis, et al. "Graph Neural Networks for Learning Equivariant Representations of Neural Networks." The Twelfth International Conference on Learning Representations.
> >
> > [3] Gelberg, Yoav, et al. "GradMetaNet: An Equivariant Architecture for Learning on Gradients." arXiv preprint arXiv:2507.01649 (2025).
> >
> > [4] Zhou, Allan, Chelsea Finn, and James Harrison. "Universal neural functionals." Advances in neural information processing systems 37 (2024): 104754-104775.
> >
> > [5] Lim, Derek, et al. "Graph Metanetworks for Processing Diverse Neural Architectures." The Twelfth International Conference on Learning Representations.
> >
> > [6] Casper, Stephen, et al. "Black-box access is insufficient for rigorous ai audits." Proceedings of the 2024 ACM Conference on Fairness, Accountability, and Transparency. 2024.

---

### Official Review · Reviewer_g7zS · 2025-10-31

**Soundness:** 3
**Presentation:** 3
**Contribution:** 3
**Rating:** 6
**Confidence:** 3

**Summary:**

The authors present an ambitious and, at first glance, quite elegant technical approach. They propose a framework in which a message-passing graph neural network learns to edit other neural networks, making them compliant with new requirements without retraining them from scratch. This is a neat trick: it’s like giving neural networks a "regulatory layer" that can adapt them to new goals on the fly.

I’m generally pleased with the conceptual elegance. It’s a clever piece of engineering and a compelling idea. However, there are a few important caveats that need to be highlighted.

First, let’s talk about the question of guarantees. The authors imply that their method can be used to achieve various goodmaking properties fairness, efficiency, data minimization, and so on but they sidestep the hardest part of the problem. The guarantees don’t come from the method itself. In fact, the method doesn’t really offer any inherent guarantees at all. It’s up to whoever is translating those goodmaking properties into mathematical form to ensure that the objectives make sense and are actually achievable. The authors effectively offload the burden of proof to someone else: they say, "Once you’ve done the hard work of turning your requirement into math, then our method applies." This is a notable limitation that's worth being explicit about: it’s a tool that requires someone else to do the heavy conceptual lifting, and it's worth avoiding the implication that mathematics alone can buy you these solutions for free.

Second, the method’s applicability is limited by its data and architecture dependence. It works for fixed architectures, which means if you want to generalize it, you have to retrain it for each new type of network. That’s a practical hurdle that limits the broader usability of the technique. It’s a nice trick, but it’s not a universal one-size-fits-all solution.

Finally, the authors bury certain limitations in footnotes, such as the assumption of a convex Pareto front. In the messy real world, multi-objective optimization often doesn’t yield such neat, convex trade-offs. This is a real limitation that should be more transparently discussed.

In conclusion, this is a clever and promising piece of work, but it comes with caveats. It’s a tool that needs careful handling and further demonstration of its generality and robustness. The authors have given us a neat technical insight, but they haven’t solved the hardest parts of the problem. That’s fine, just means there’s more work to be done.

**Strengths:**

- Elegant formulation of model editing as a learnable mapping in weight space.
- Uses graph metanetworks to perform edits in a single inference step: computationally efficient in a sense.
- Unifies multiple compliance tasks (fairness, pruning, data minimisation) under one objective.
- Demonstrates consistent Pareto improvements over post-hoc baselines.

**Weaknesses:**

- Depends on fixed architectures; generalization across model families untested.
- Offloads hardest work theoretical work (formalising requirements) onto the user (may not be a weakness per se)
- Assumes convex Pareto fronts and differentiable requirement objectives; necessary move perhaps, no shame in just saying "here is a nice technical trick, let's not worry about guarantees."
- “Regulatory compliance” rhetoric overstates what is actually a tuning heuristic.

**Questions:**

- How sensitive is the learned editor to the distribution of training networks? Does the universe conspire in our favour to make all editors behave roughly the same?
- Can metanetworks be stacked or composed for multiple simultaneous requirements? Sure there are no theoretical guarantees but does it work in the wild?
- What safeguards prevent catastrophic edits or hidden performance degradation? Is there a meta-meta-network for that?

---

> ### Author Response · Authors · 2025-11-26
> **Rebuttal by Authors**
>
> We sincerely thank Reviewer g7zS for their thorough and insightful review. We greatly appreciate that they engaged deeply with our work and raised thought-provoking points that motivated us to evaluate our framework in new directions (metanetwork composition, additional multi-objective methods) while also helping us clarify and expand upon important sociotechnical considerations.
>
> > Weakness 1: *Depends on fixed architectures; generalization across model families untested.*
>
> We appreciate the reviewer's attention to this important aspect of our work. However, we would like to respectfully clarify a potential misunderstanding regarding architectural diversity in our experiments. Our method does ***not*** depend on fixed architectures. The neural network dataset $\mathcal{D}_m$​ that we construct for training the metanetwork comprises models with substantial architectural variation. As detailed in Section A.6 of the appendix, we sample MLP architectures with **varying depth, width, skip connections** (between any tuple of layers), and **many more hyperparameters**.
>
> The reviewer is correct, however, that our evaluation is limited to the MLP family. Although the Graph Metanetwork paradigm [1] enables training on a dataset of different model families, by injectively mapping NNs to graphs, we have not yet evaluated our method is such use cases. This is indeed a limitation that we acknowledge in our manuscript, and extending our framework to diverse model families poses an important direction for future work.
>
> > Weakness 2: *Offloads hardest work theoretical work (formalising requirements) onto the user (may not be a weakness per se)*
>
> We thank the reviewer for raising this point. We would like to emphasise that *the mathematical formulation of diverse compliance requirements is a core contribution of our work on its own sake* . Since requirements defined in legal documents or government reports typically lack mathematical rigor, translating them into mathematical objectives is essential. Notably, we identify this translation as the responsibility of the academic and research community rather than of end-users. We envision that our work can function as a step towards the initiation and systematisation of this process and motivate future research in this direction.
>
> > Weakness 3: *Assumes convex Pareto fronts and differentiable requirement objectives*
>
> **Convex Pareto front**
> We thank the reviewer for this insightful observation, which motivated us to investigate more modern multi-objective optimization methods suitable for non-convex problems. In response to this feedback, we conducted additional experiments using the Exact Pareto Optimal (EPO) Search algorithm [2] in place of Linear Scalarisation (LS) for the fairness objective. Unlike LS, EPO makes no assumptions regarding the convexity of the Pareto front and is capable of identifying exact solutions for any given preference vector.
>
> Please refer to Figure [9] in the appendix of the revised manuscript. From our experiments, we observe that GMN - EPO yields a Pareto front qualitatively similar to that obtained with GMN - LS. However, GMN - EPO covered a substantial portion of the Pareto front with relatively few preference vectors, unlike LS, which requires more extensive exploration. Notably, our experiments evaluate *model generalization to unseen data points*, not just Pareto coverage. Consequently, while EPO guarantees exact Pareto optimal solutions during training, this optimality is not guaranteed on validation and test splits.
>
> **Differentiability**
> As the metanetwork is optimized through gradient-based methods, differentiability constitutes a fundamental requirement. Nevertheless, as explicitly detailed in the definition of each requirement in section 3.2, all of the discussed requirements are formulated inevitably as *non-differentiable objectives*. In the cases of Data Minimization and Pruning, predicting a hard mask on the nodes (former) or the edges (latter) is not differentiable, while for the Fairness case, requirement objective relies on the predicted hard labels of the edited model, which again hinders the differentiability.  To address the non-differentiability challenge, in this paper, we introduce continuous relaxations that enable gradient-based optimization, but other approaches (e.g. reinforcement learning) can be explored as alternatives.
>
> > Weakness 4: *“Regulatory compliance” rhetoric overstates what is actually a tuning heuristic.*
>
> We acknowledge that the term 'compliance' may confuse the reader and will refine our wording in the revision. We respectfully disagree, however, with characterizing our approach as a 'heuristic.' Our method is grounded in a multi-objective framework that precisely translates our goal into a mathematical formulation, which we then solve in a data-driven fashion via the metanetwork. This transcends simple heuristics such as feature selection in data minimisation or magnitude-based pruning in neural networks.

---

> ### Author Response · Authors · 2025-11-26
> **Rebuttal by Authors**
>
> > Question 1: *How sensitive is the learned editor to the distribution of training networks? Does the universe conspire in our favour to make all editors behave roughly the same?*
>
> **Architectural Diversity**: As clarified in our response above, our training dataset $\mathcal{D}_m$ comprises MLPs with substantial architectural variation, while our graph metanetwork generalizes effectively across these diverse architectures.
>
> **Task Distribution**: A key assumption in our current framework is that all models within $\mathcal{D}_m$ are trained on the same task distribution $p_d$, as discussed in out limitations. Extension to multi-task scenarios, where the metanetwork could edit models trained on diverse tasks, is discussed in our response to Reviewer 4ehz and represents an important direction for future work.
>
> We would be grateful if the reviewer could provide additional clarification regarding their reference to 'distribution sensitivity' in this comment, as this would enable us to respond more precisely to their concern.
>
> > Question 2: *Can metanetworks be stacked or composed for multiple simultaneous requirements?*
>
>
>
> We thank the reviewer for proposing this interesting experiment. We evaluate the composition of a metanetwork $F_{dm}$ trained on Data Minimization with a metanetwork $F_{pr}$ trained on Pruning, to obtain $F = F_{pr} \circ F_{dm}$. To do so, we use a set of metanetworks trained on various $\lambda$ values for each objective, $\Lambda_{dm}$ and $\Lambda_{pr}$ respectively, and evaluate the composition of $\Lambda_{dm} \times \Lambda_{pr}$ metanetworks.
>
> We plot our results in Figure 8 in the appendix of the updated manuscript. For reference, we compare with the composition of the best baselines from the Data Minimization and Pruning experiments and we observe that our method fully dominates the Pareto fronts that occur, even when compared to the best alternatives.
>
>
> *The outcome of this experiment functions as additional empirical evaluation to our framework*: the fact that metanetworks trained independently can be composed effectively suggests that each metanetwork preserves the parameter distribution sufficiently well that subsequent metanetworks can operate on their outputs. In other words, this is empirical evidence that our preservation objective (minimizing JSD) preserves the NNs' functionality without causing a harmful side-effect (distribution shift).
>
> > Question 3: *What safeguards prevent catastrophic edits or hidden performance degradation? Is there a meta-meta-network for that?*
>
>
> Again, we thank the reviewer for their constructive question. We acknowledge that such a discussion was missing from the original manuscript, thus we have added our thoughts on human oversight and the biases the metanetwork itself may carry, as these considerations are essential for any work with sociotechnical implications.
>
> In this work we proposed a framework for automated model editing for requirement compliance. However, such methods ***must not bypass human oversight***. While our framework provides interpretable Pareto fronts that facilitate informed decision-making, human judgment remains essential. Domain experts and practitioners must validate edited models, assess trade-offs in their specific deployment context, and make the final decision regarding deployment, in a human-in-the-loop regime. This limitation applies broadly to automated compliance tools and represents an important area for future work in AI governance.
>
>
> Regarding hidden performance degradation, our choice of Jensen-Shannon Divergence as the preservation metric is specifically motivated by the need to minimize functional deviation from the original model. JSD provides a bounded, interpretable measure of how much the model's output distribution has changed, serving as a primary safeguard against catastrophic edits that would fundamentally alter model behavior. As always, beyond JSD, practitioners can monitor additional metrics (e.g., accuracy, F1 scores) on validation sets before deployment to detect any unintended degradation.
>
>
> ---
>
> [1] Lim, Derek, et al. "Graph Metanetworks for Processing Diverse Neural Architectures." The Twelfth International Conference on Learning Representations.
>
> [2] Mahapatra, Debabrata, and Vaibhav Rajan. "Multi-task learning with user preferences: Gradient descent with controlled ascent in pareto optimization." International Conference on Machine Learning. PMLR, 2020.

---

> > ### Comment · Reviewer_g7zS · 2025-11-26
> > **Score increased**
> >
> > Yes, this addresses my substantial concerns very happily. Thank you very much: moving to clear accept.

---

### Official Review · Reviewer_RyXd · 2025-10-31

**Soundness:** 3
**Presentation:** 3
**Contribution:** 2
**Rating:** 4
**Confidence:** 4

**Summary:**

The paper proposes a metanetwork that edits trained models in a single pass to meet requirements (data minimization, fairness, sparsity), replacing intensive optimization loops. Framing compliance as a learned weight-space mapping is well-motivated.

**Strengths:**

It's good to see the evaluation of how robust the edited models remain when the underlying data distribution (pd) shifts.

**Weaknesses:**

The regulatory positioning could be more precise by linking edits to specific AI Act or ISO/IEC risk-management processes. The relation to unlearning is better reframed as complementary rather than alternative, since this method edits parameter rather than removing data influence.

**Questions:**

The paper doesn’t explain how λ is chosen or how sensitive results are to it. Specify the sampling strategy for the Dd subset.

---

> ### Author Response · Authors · 2025-11-26
> **Rebuttal by Authors**
>
> > Weakness 1: *The regulatory positioning could be more precise by linking edits to specific AI Act or ISO/IEC risk-management processes.*
>
> We thank the reviewer for this constructive suggestion. Below, we clarify how each of our three NN edits directly relates to regulatory requirements:
>
> **Data Minimization**: Explicitly mandated by Article 5(1)\(c) of the EU GDPR, which verbatim states that *"Personal data shall be adequate, relevant and limited to what is necessary in relation to the purposes for which they are processed"*. This requirement is similarly reflected in Section 1798.100(a)(1)–(2) of the California Privacy Rights Act (CPRA) and Article 6 of Brazil’s General Personal Data Protection Law (LGPD). Moreover, ISO 42001’s governance framework includes data governance as a core objective. One of its risk controls is data inventory and minimization.
>
> **Bias Mitigation**: This requirement aligns with Article 10 of the EU AI Act, which mandates that high-risk AI systems ensure appropriate levels of accuracy and avoid discrimination. The Equalized Odds fairness metric used in our method—though not the only metric applicable—promotes equal true positive and false positive rates across protected groups, thereby directly supporting the Act’s non-discrimination obligations.Additionally, this requirement relates to ISO/IEC 42001:2023 AI management system requirements and ISO/IEC 23894:2023 AI risk management guidance.
>
> **Pruning**: Although not explicitly mandated, computational efficiency, the utmost goal of pruning, is a widely recognized and ubiquitous requirement across domains such as embedded systems, edge and mobile computing, IoT, real-time analytics, and large-scale machine learning.
>
> > Weakness 2: *The relation to unlearning is better reframed as complementary rather than alternative, since this method edits parameter rather than removing data influence.*
>
> We appreciate the reviewer's careful reading. However, we would like to clarify a potential misunderstanding. In our manuscript, we refer to unlearning as only an example among several existing application-specific NN editing methods, in the Related Work section. We do **not** position our method as an alternative specifically to unlearning, nor do we claim to address data removal or the specific objectives of unlearning.
>
>
> > Question 1: *The paper doesn’t explain how λ is chosen or how sensitive results are to it.*
>
> We thank the reviewer for this important question. In our framework, to solve the multi-objective problem we employ linear scalarisation by using a weighting coefficient λ>0. We stress here that λ reflects a user-defined preference between the two objectives. Consequently, we do not intend to select a *single* λ. Instead, we systematically vary it to explore the Pareto front and obtain the best possible trade-off for each preference parameter. As explained in Section 5 of the manuscript, we conduct a hyperparameter search, for several values of λ, and select the models that lie on the Pareto front of the *validation set*.
>
>
>
> > Question 2: *Specify the sampling strategy for the D_d subset.*
>
> As described in the experimental setup in Section 5 of the manuscript, $\mathcal{D}_m$ is defined as the dataset of models on which the metanetwork will be trained on and as $\mathcal{D}_d$ the task dataset on which each model of $\mathcal{D}_m$ was trained on. To construct datasets applicable to all three requirements, in our experiments we selected the tabular datasets *Adult* and *Bank Marketing* from the UCI repository. Thus, $\mathcal{D}_d$ corresponds to one of these datasets without any additional sampling.
>
> During metanetwork training, we rely exclusively on the validation split of D~d~ datasets, as required by both the Preservation and the Requirement (only in the Fairness use case) Objective, as shown in equations (1) and (2) respectively. We further restrict ourselves to a subset of the validation split to avoid assuming access to data samples that the input model has already seen.
>
> Finally, a sampling strategy was only used to construct a $\mathcal{D}_m$ of varying MLP architectures, as detailed in Section A.6 of the appendix.

---

### Author Response · Authors · 2025-11-26

We would like to thank the reviewers for their thorough evaluation of our paper and their constructive feedback, which helped us improve our empirical evaluation to further corroborate our claims and identify potential future directions. In the following comments, we gather the strengths pointed out by the reviewers and summarise our rebuttal response and changes that will be made in an updated version of the manuscript.

We appreciate that our contribution was recognized across multiple dimensions:

### **Problem Formulation & Motivation**
* **Well-motivated and timely contribution** as characterized by Reviewers **RyXd** (*"[...]is well-motivated"*), **4ehz** (*"The motivation is strong and well-timed"*) and **eYmo** (*"Motivation is well-argued "*).
* **Ambitious idea and elegant formulation** as characterised by Reviewer **g7zS** ("ambitious", *"pleased with the conceptual elegance."*, *"a compelling idea"*, *"Elegant formulation"*).
* **On a gap in a pressing topic** as emphasized by Reviewer **4ehz** (*"a pressing issue in trustworthy AI"*, *"[...] highlighting a gap in flexible, general-purpose model editing"*.).
* **Clear presentation** as pointed out by Reviewer **eYmo** (*"the paper is clearly written, with a crisp problem setup and method description."*).

###  **Unifying Framework & Architecture**
* **Technical and conceptual elegance** as characterised by Reviewer **g7zS** ("quite elegant technical approach.", *"clever piece of engineering"*, *"a neat trick"*).
* **Flexibility of framework** as highlighted by Reviewers **g7zS** (*"Unifies multiple compliance tasks [...] under one objective."*) and **eYmo** (*"Experiments are thoughtfully structured across diverse requirements [...], which convincingly demonstrate versatility rather than one task."*, *"Covers three compliance needs"*).
* **Novelty of method** as acknowledged by Reviewer **4ehz** (*"The metanetwork architecture is innovative: [...] is a cutting-edge idea that leverages recent advances in meta-learning and symmetry-preserving design."*).
* **Generalisation on varying architectures** as pointed out by Reviewer **eYmo** (*"Accommodates different parameter shapes [...], rather than assuming a fixed MLP architecture."*).



### **Empirical Performance & Computational Efficiency**
* **Computational Efficiency** of our method highlighted by Reviewers **g7zS** (*"edits in a single inference step: computationally efficient in a sense."*) and **eYmo** (*"extremely fast inference"*).
* **Performance** acknowledged by Reviewers **eYmo** (*"promising Pareto trade-offs"*) and **g7zS** (*"Demonstrates consistent Pareto improvements"*).


## Rebuttal Summary
We thank the reviewers that motivated us to further expand our empirical evaluations on four different directions:

### **Training set size ablation**

We conducted an ablation study on the training set size, as pointed out by reviewers **4ehz** and **eYmo**,  focusing on the pruning task. (Figure 4)

**Results**: The metanetwork exhibits strong sample efficiency across different training set sizes. Even with only **10%** of the training data, our method outperforms all baselines except at edge cases. At **25% of the data, GMN substantially outperforms all baselines across the entire Pareto front**.

### **Composing metanetworks**
Motivated by Reviewer **g7zS**, we evaluate the composition of a metanetwork $F_{dm}$ trained on Data Minimization with a metanetwork $F_{pr}$ trained on Pruning, to obtain $F = F_{pr} \circ F_{dm}$. (Figure 8)

**Results**:  Our composed metanetworks fully dominate the Pareto front, significantly outperforming the composition of the best baselines. These results stand as an empirical evidence that our *preservation objective preserves the NNs' functionality without introducing a harmful side-effect (distribution shift)*.

### **Exact Pareto Optimal Search**
Motivated by Reviewer **g7zS**, we evaluated an alternative multi-objective optimization method, which does not assume convexity and can identify exact solutions. (Figure 9).

**Results**: EPO algorithm enabled us to cover a substantial portion of the Pareto front with relatively few preference vectors, unlike LS, which requires more extensive exploration.


### **Human oversight**
Motivated by Reviewer **g7zS**, we included a discussion in the updated manuscript on human oversight where we emphasize that complete automation is beyond our intended scope, particularly in sociotechnical contexts where human judgment remains essential.

---

### Meta-Review · Area_Chair_nHAX · 2026-01-07

**Summary:**

This paper proposes a meta-network that learn to edit trained models (specifically MLPs) to satisfy requirements (e.g. fairness, pruning, etc). The key idea is to formulate the requirement as a multi-objective optimization problem to balance performance + constraints and train a graph meta-network. Generally, the reviewers agree the problem is important and the idea is interesting. However, some shared concerns are not well-addressed, including the practicality of the proposed method which only works for MLP models and tabular data. Hence a rejection is recommended.

**Reviewer Concerns:**

* Reviewer g7zs's concern on fixed architecture is still outstanding
* Reviewer 4ehz's concern on limited scope to MLP and tabular data, not scalable to larger models in text and vision is still outstanding
* Reviewer 4ehz's concern on training cost analysis is partially addressed
* Reviewer 4ehz's concern on interpretability analysis is still outstanding
* Reviewer eYmo's concern on generalization ability and data dimensionality is still outstanding

**Reviewer Scores:**

* Reviewer RyXd may remain score or increase to 5
* Reviewer g7zs stated substantial concerns addressed and will increase score to 8
* Reviewer 4ehz likely to remain score as 4 or downgrade to 3
* Reviewer eYmo stated concern are somehow addressed though generalization ability and data dimensionality are not completely resolved, and remain score of 6 but increase confidence to 3

---

### Decision · Program_Chairs · 2026-01-26

Reject